# Niosomes as Vesicular Carriers: From Formulation Strategies to Stimuli-Responsive Innovative Modulations for Targeted Drug Delivery

**DOI:** 10.3390/pharmaceutics17111473

**Published:** 2025-11-14

**Authors:** Andra Ababei-Bobu, Bianca-Ștefania Profire, Andreea-Teodora Iacob, Oana-Maria Chirliu, Florentina Geanina Lupașcu, Lenuța Profire

**Affiliations:** 1Department of Pharmaceutical and Therapeutic Chemistry, Faculty of Pharmacy, Grigore T. Popa University of Medicine and Pharmacy of Iași, 16 University Street, 700115 Iași, Romania; andra-ababei@email.umfiasi.ro (A.A.-B.); andreea.panzariu@umfiasi.ro (A.-T.I.); oana-maria.ionescu@umfiasi.ro (O.-M.C.); lenuta.profire@umfiasi.ro (L.P.); 2Department of Internal Medicine, Faculty of Medicine, Grigore T. Popa University of Medicine and Pharmacy of Iasi, 16 University Street, 700115 Iași, Romania; bianca-stefania.profire@umfiasi.ro

**Keywords:** niosomes, non-ionic surfactant vesicles, smart stimuli-responsive nanocarriers, targeted drug delivery system

## Abstract

Niosomes (NIOs), a class of nanovesicular drug delivery system, have garnered significant attention due to their unique architecture, resulting from the self-assembly of non-ionic surfactants (with or without cholesterol) in aqueous media. This bilayered structure enables the encapsulation of both hydrophilic agents in the aqueous core and lipophilic compounds within the lipid bilayer, offering remarkable versatility in therapeutic applications. This article provides an overview of the key principles underlying niosomal formulations, including their composition, preparation methods, formulation conditions and the critical physicochemical parameters influencing vesicle formation and performance. Special emphasis is placed on recent innovations in surface and content modifications that have led to the development of stimuli-responsive niosomal systems, with precise and controlled drug release. These smart carriers are designed to respond to endogenous stimuli (such as pH variations, redox gradients, enzymatic activity, or local temperature changes in pathological sites), as well as to exogenous triggers (including light, ultrasound, magnetic or electric fields, and externally applied hyperthermia), thereby enhancing therapeutic precision. These surface and content modulation strategies effectively transform conventional NIOs into intelligent, stimuli-responsive platforms, reinforcing their innovative role in drug delivery and highlighting their significant potential in the development of smart nanomedicine.

## 1. Introduction

Pharmaceutical nanocarriers, a fundamental component of nanomedicine, provide an innovative approach to drug delivery, addressing formulation challenges and enhancing the physicochemical characteristics of diverse pharmacological agents [1,2]. Compared to conventional systems, they enable precise control of drug release and targeted delivery, thereby improving pharmacokinetics, biodistribution, bioavailability, and the active concentration at the pathological site [3,4]. Among the various nanocarriers developed to date (solid lipid nanoparticles, polymeric micelles, dendrimers, polymeric nanoparticles, inorganic nanoparticles), vesicular drug delivery systems (VDDSs) have shown notable effectiveness in enhancing the bioavailability of various Active Pharmaceutical Ingredients (APIs), representing a promising nanotechnology-based approach [3,5,6]. Within VDDSs, niosomes (NIOs) and proniosomes have attracted particular interest for their proven therapeutic efficacy, surpassing liposomes through superior physicochemical attributes, including enhanced chemical and thermal stability, reduced formulation costs, and improved biocompatibility and biodegradability [7,8].

NIOs represents a specific nanovesicular system, initially documented in the 1970s by researchers at L’Oréal (Clichy, France), who utilized them for cosmetic purposes, subsequently launching the product under the name Niosôme by Lâncome in 1986 [9]. Since then, they have been proposed for various applications in the domains of food, pharmaceuticals, and cosmetic sciences [9]. NIOs possess a distinctive architecture resulting from the self-assembly of non-ionic surfactants (NISs) with cholesterol (Chol) in aqueous environments, resulting in formation of uni- or multi-lamellar vesicles capable of encapsulating both hydrophilic and hydrophobic agents [10]. The incorporation of Chol within the lipid bilayer enhances membrane stability, increases drug entrapment efficiency (EE), and modulates vesicle permeability, properties that support their superior biocompatibility, controlled release behavior, and overall suitability for prolonged therapeutic applications [11].

The incorporation of NISs not only promotes the structural organization of NIOs, through interactions within lipid bilayers, but also extend their circulation time, thereby improving therapeutic efficacy and facilitating targeted drug delivery [12,13].

Figure 1 illustrates the bilayered vesicular structure of niosomes, self-assembled from amphiphilic molecules, capable of encapsulating hydrophilic agents within the aqueous core and lipophilic compounds within the lipid bilayer. Their high surface-to-volume ratio and amphiphilic nature enable modulation of the physicochemical properties and bioactivity of loaded APIs, thereby improving solubility and stability [14,15].

Researchers have recently focused on fully exploring the advantages offered by niosomal formulations to enhance the solubility and bioavailability of poorly soluble APIs. These systems contribute to reducing the required dosage and enhancing both photo- and chemical-stability [16], protecting encapsulated agents against degradation mechanisms: proteolytic enzyme activity [17] or photo-induced oxidation [18].

Moreover, the versatility of niosomal carriers has been extensively investigated for the delivery of a wide range of therapeutic molecules, including proteins [19], peptides [20,21,22], antigens [23,24], genetic material [10,25,26,27], antibodies [28], and hormones [29,30,31,32].

This article provides an overview of NIOs as VDDSs, addressing the key principles underlying niosomal, including composition, preparation methods, formulation conditions, and the physicochemical parameters influencing vesicle formation and performance. Furthermore, it highlights surface and content modulation strategies for the development of stimuli-responsive systems for targeted drug delivery, emphasizing their role in formulation innovation and in transforming NIOs into smart nanomedicine platforms with possible clinical applications. The review also points out the most advanced preclinical of stimuli-responsive NIOS and the current state of clinical translation as well as the major challenges in clinical development.

## 2. Formulation Aspects of NIOs

### 2.1. Methods of NIOs Preparation

NIOs can be classified into multiple categories according to their structural characteristics, functional derivatization, preparation methods, nature of the encapsulated APIs, and the type of surfactants employed [33,34], as summarized in Table 1. A subclassification, based on the nature of the NISs and additional components incorporated during fabrication, which includes ethosomes [35], bola-surfactant NIOs [36], transfersomes [37], discomes [38,39], aspasomes [40,41], elastic NIOs [42,43] and polyhedral NIOs [44], with their principal features, are also provided in Table 1.

NIOs provide significant benefits, including enhanced chemical stability, extended shelf-life without the need for special storage conditions, improved bioavailability of poorly soluble drugs, and adaptable methods of preparation [45,46]. By adjusting preparation parameters (type and ratio of additives or their association), both drug entrapment efficiency and vesicular characteristics (composition, fluidity, shape, size, surface charge, and lamellarity) can be precisely modulated [47], while additional functionality may be conferred through surface modification with ligands (carbohydrates, glycoproteins, etc.) anchored to surfactant head groups [11,48].

Each method for NIOs preparation presents inherent advantages and limitations. In general, NIOs fabrication is straightforward, cost-effective, and suitable for large-scale production due to the low price of NISs. However, certain methods like proniosome technology may require advanced equipment, thereby increasing costs [49]. Specific incompatibilities associated with some methods have been documented: the “bubble” method is prone to instability during prolonged storage; the ether injection and reverse-phase evaporation methods may lead to toxicity from residual organic solvents; the transmembrane pH gradient method often suffers from low reproducibility and poor standardization; the heating method is incompatible with thermosensitive APIs [50,51,52]. In summary, the benefits of niosomal formulations surpass their drawbacks, as illustrated in Table 2.

### 2.2. Formulation Factors Affecting NIOs Characteristics

#### 2.2.1. Non-Ionic Surfactant (NISs)

NISs represent the main compounds used in NIOs preparation due to their superior ability to target and sustain drug delivery. Compared with anionic, amphoteric, or cationic surfactants, they offer better compatibility and stability while generally exhibiting lower toxicity [53]. In terms of cell surface interactions, NISs preserve a near physiological pH, proving reduced toxicity, hemolytic activity, and irritation [54]. They are amphiphilic molecules characterized by a hydrophobic tail and a hydrophilic head group, which enable the formation of two distinct regions with differing solubility. A broad spectrum of such surfactants has been employed in NIOs formulation, including alkyl ether and alkyl ester derivatives, glucosyl alkyl ethers, pluronic-type block copolymers, as well as fatty alcohols and fatty acids. These NISs are briefly illustrated in Figure 2 [33,34].

*Alkyl ethers surfactants* can be categorized into two subgroups according to the nature of their hydrophilic head groups: alkyl glyceryl ethers, with hydrophilic head groups derived from glycerol, and polyoxyethylene alkyl ethers (Brij series), which contain ethylene oxide subunits head groups. Generally, ether-based surfactants are preferred over ester analogs, which are rapidly degraded and rendered unstable under in vivo conditions [55]. Additional features supporting their suitability for NIOs preparation include low allergenic potential for skin administration, the ability to form stable vesicles capable of encapsulating macromolecules such as proteins and peptides, and enhanced compatibility, when combined with other surfactants. While alkyl glyceryl ethers have been employed in the delivery of agents such as methotrexate [56] and 5,6-carboxyfluorescein [57], members of the Brij series have been particularly applied in the oral delivery of insulin, effectively preventing its inactivation by gastric juice [31].

*Alkyl ester derivatives* are a class of NISs frequently employed in the successful formulation of NIOs, mostly due to their non-irritant and non-toxic properties. Current research on these surfactants aims to establish a correlation between drug EE and release kinetics in relation to differences in alkyl chain length and head group [58]. Sorbitan fatty acid esters (Spans) are among the most favored NISs in formulation of NIOs. Studies have demonstrated that the EE of NIOs is strongly influenced by the alkyl chain length of the Span surfactant and the unsaturation grade. Accordingly, Span 60, which contains a longer saturated C18 chain, produces NIOs with higher EE compared to shorter chain analogs such as Span 40 (C16) [59,60]. For example, in a study on fluconazole-loaded NIOs, the EE was shown to decrease in the order Span 60 (C18) > Span 40 (C16) > Span 20 (C12) > Span 80 (C18). These data reflect that Span 20, Span 40, and Span 60 share an identical hydrophilic head group but differ in their alkyl chain length; longer saturated chains, as in Span 60, enhance bilayer rigidity and reduce permeability, thereby improving EE. In contrast, shorter chains (Span 20 or Span 40) or the presence of unsaturation (Span 80) lead to less ordered and compact bilayers and consequently lower drug encapsulation [61].

Tween surfactants (polysorbates) are polyoxyethylene sorbitan fatty acid esters, structurally defined by a hydrophilic head group composed of oligo(ethylene glycol) (OEG) chains and a hydrophobic tail derived from fatty acid esters. Their dual structural characteristics, namely the presence of relatively long alkyl chains together with a significant hydrophilic moiety, make them particularly suitable for encapsulating hydrophilic drugs [58]. EE within the tween series generally decreases in the order Tween 20 (C12) > Tween 60 (C18) > Tween 40 (C16) > Tween 80 (C18), highlighting an inverse relationship between alkyl chain length and drug-loading capacity. In this context, Tween 80, which incorporates a longer but unsaturated C18 chain, exhibits the lowest drug EE [62]. Beyond EE, the presence of hydrophilic polyoxyethylene (POE) chains in Tweens confers additional functional properties to the resulting NIOs. For example, Tween 80-based NIOs have demonstrated notable performance in gene delivery and transfection efficiency, whereas Tween 20-based NIOs exhibited structural and compositional features that facilitated their transport across Caco-2 cell monolayers, thereby enhancing drug permeation through the intestinal epithelial barrier and improving therapeutic outcomes [63].

*Pluronic triblock copolymers* (poloxamers) constitute a class of amphiphilic copolymers composed of hydrophilic ethylene oxide (EO) and hydrophobic propylene oxide (PO) units. They exhibit a characteristic triblock arrangement, with a central polypropylene oxide (PPO) block flanked by polyethylene oxide (PEO) segments of equal length on either side [64,65]. These linear EO–PO–EO NISs have recently been employed in NIOs formulation for various applications, including the injectable delivery of brucine [66] and the development of thermo-responsive azithromycin-loaded niosomal gel [67]. Furthermore, mixed-surfactant systems combining sorbitan monolaurate with poloxamer 184 demonstrated superior performance in the oral delivery of diacerein, achieving high EE with lower cholesterol requirements [68].

*Glucosyl alkyl ethers* constitute a class of surfactants that includes both glucosides and alkyl poly-glucosides, compounds with pronounced amphiphilic properties enabling the formation of stable vesicles. In the glucoside series, the glucosides of myristyl-, cetyl- and stearyl-alcohols have been shown to form vesicular structures, whereas those of shorter-chain alcohols such as decyl-, lauryl-, and octyl, did not [69]. The sugar-based alkyl polyglucosides have attracted increasing interest as sugar moieties can substitute the EO units that usually form the polar region of conventional surfactants, whose use in food and pharmaceutical industries has declined due to concerns about toxic potential. Due to their favorable profile of rapid biodegradability and low toxicity, glucosyl alkyl ethers have increasingly replaced EO-based surfactants in various applications. Studies support their advantageous dermatological properties in cosmetology, as well as their in vitro ability to facilitate cutaneous tretinoin delivery from mixt vesicles formulations (octyl-decyl-polyglucoside and decyl-polyglucoside), achieving enhanced transdermal and cutaneous delivery [70,71].

*Bola and Gemini NISs* represent specific class of ether-based surfactants. Bola amphiphiles are characterized by the presence of two hydrophilic head groups, whereas Gemini surfactants possess both dual head groups and dual hydrophobic tails [72,73]. More specifically, bola surfactants (also termed bolaphiles or α,ω-type surfactants) are amphiphilic molecules consisting of hydrophilic groups at both ends of a long hydrophobic hydrocarbon chain [74]. Compared with single-headed surfactants, they exhibit higher water solubility, increased critical micelle concentration (CMC), and lower aggregation, properties that derive from the presence of the second hydrophilic head group [75]. By contrast, Gemini surfactants, are dimeric molecules structurally related to bola amphiphiles, comprising two hydrophobic tails and two hydrophilic heads connected by a spacer located at or near the head groups [76]. In comparison to their monomeric analogs, they display reduced CMC, enhanced micelle formation and solubilization capacity, and improved stability, while maintaining non-irritant, non-toxic, and non-hemolytic profiles [77].

*Fatty alcohols and fatty acids*, as simple single-chain amphiphiles, can assemble into bilayer membrane vesicles, influencing the size, stability, and EE, while entrapping both macromolecules and small molecules [78]. While fatty alcohol-based vesicular systems (stearyl-, cetyl-, and myristyl-alcohol) typically form small multilamellar vesicles with controlled release [79], fatty acid-based systems (stearic-, palmitic-, myristic-, oleic-, linoleic-, octanoic-, and decanoic-acids) generate larger multilamellar vesicles. The formation of these fatty acid-based vesicles is strongly influenced by pH, as bilayer membranes occur spontaneously when the pH is close to the pKa of the bilayer-incorporated fatty acid, whereas at higher pH micelles predominate and at lower pH oil droplets condense [80]. The stability of these bilayers at pH values near the pKa is attributed to hydrogen bonding between adjacent protonated and ionized carboxylates, which reduces electrostatic attraction between vicinal head groups, a mechanism further supported by the greater stability of vesicles composed of fatty acid and fatty alcohol mixtures compared with those formed solely from fatty acids [78].

**Key surfactant characteristics**, including the hydrophilic–lipophilic balance (HLB) value, the critical packing parameter (CPP), and the gel-liquid crystalline transition temperature (Tc), play a decisive role in promoting the formation of bilayered vesicles rather than micelles [55].


*The Hydrophilic–Lipophilic Balance*


HLB is a dimensionless parameter, whose value expresses the ratio of the non-ionic surfactant’s hydrophilic and lipophilic content, a direct indicator of surfactant solubility. Since HLB can influence the formulation of NIOs from several perspectives, it is considered to calculate the HLB value of the entire content, based on all the surfactants and their concentrations used [55].

One of the most critical aspects is its impact on drug EE, which is also affected by the alkyl chain length of the surfactant. Several studies have demonstrated that lower HLB values are generally associated with increased EE and improved stability. For example, in Span–Chol-based NIOs, acetazolamide [59] and aceclofenac [60] exhibited higher EE when formulated with Span 60 (HLB 5) compared to Span 40 (HLB 6.7). However, subsequent investigations have highlighted that this relationship is not universally valid. Ruckmani et al. [62] reported different findings for zidovudine EE, achieving values of 83.8% with Tween 20 (HLB 16.7), 82.4% with Tween 60 (HLB 14.9), 81.6% with Tween 40 (HLB 15.6), and 79.5% with Tween 80 (HLB 15). In this case, drug EE correlated more closely with the alkyl chain length, which also influences the HLB. Specifically, longer alkyl chains (Tween 80 (C18) > Tween 60 (C18) > Tween 40 (C16) > Tween 20 (C12)) were associated with reduced drug entrapment.

With respect to NIOs size, higher HLB values, resulting from longer alkyl chain lengths, have generally been associated with the formation of larger vesicles. Within the Span series, for example, the smallest average vesicle sizes were reported with Span 80 (HLB 4.3) (0.319 μm) compared to Span 20 (HLB 8.6) (0.448 μm) [61], and with Span 60 (HLB 4.7) (~100 nm) compared to Span 40 (HLB 6.7) (~230 nm) [81]. This phenomenon can be attributed to the reduced number of hydrophilic groups, which decreases surface-free energy as hydrophobicity increases, thereby favoring the formation of smaller particles [82,83]. A similar trend has been observed in the Brij series, which contains surfactants with polyoxyethylene head groups and variable alkyl chain lengths. In this case, smaller vesicles were obtained with Brij 72 (HLB 4.9) (0.287 μm) compared to Brij 52 (HLB 5.3) (0.408 μm) and Brij 76 (HLB 12.4) (0.443 μm) [61].

From a practical standpoint, the literature generally indicates that stable vesicular systems can be formed within a specific HLB range, although these limits may vary slightly depending on formulation conditions. An HLB range of 4–8 is typically considered optimal for generating stable NIOs [84], whereas values between 14 and 17 are deemed unsuitable for NIOs formation [85]. For surfactants with HLB values above 6, the addition of Chol is necessary to promote bilayer formation and membrane stabilization. This effect is attributed to interactions between the 3-OH group of Chol and the hydrophobic tail of the surfactant adjacent to its polar head group, thereby enhancing hydration and structural integrity of the vesicles [54,85].


*Critical Packing Parameter*


The nominal geometric features of an amphiphilic surfactant molecule can be described by CPP, which predicts its vesicle-forming ability, the type of aggregation obtained, and the morphology of the resulting self-assembled structures [83]. CPP is a dimensionless parameter defined by the relationship between the hydrophilic head group area at the lipid–water interface (*A_0_*), the hydrophobic chain volume (*V*), and the hydrophobic chain length (*l_0_*) (Figure 3) [86,87], according to the equation:$$\mathrm{CPP}\;=\;\frac{V}{l_{0}\;\times \;A_{0}}$$
where *V* represents the hydrophobic chain volume (Å^3^), *l_0_*, the hydrophobic chain length (Å), and *A_0_*, the hydrophilic head group area (Å^2^).

CPP provides insight into the formation of surfactant aggregates, which are governed by the molecular symmetry of the surfactant, as summarized in Table 3 [88]. It should be noted, however, that CPP values often provide only approximate guidance. Although the parameters *V*, *l_0_*, and *A_0_* are generally considered constant for a given surfactant, they may vary with changes in solution conditions such as temperature or salinity. Such variations can induce significant phase transitions, which in turn may hinder the successful formation of NIOs.


*Gel-Liquid Transition Temperature*


The gel-liquid phase transition temperature is a defining characteristic of amphiphilic molecules, including both surfactants and lipids. Due to their amphiphilic nature, these molecules self-assemble into bilayer structures whose properties are strongly influenced by composition and temperature. Below Tc, the bilayer adopts a gel phase (Lβ) (Figure 4a), in which the amphiphilic molecules are arranged in a solid-ordered state. Above Tc, the bilayer undergoes a transition to the liquid-crystalline phase (Lα), also referred to as the liquid-disordered state (Figure 4b) [89,90]. This temperature-dependent phase behavior reflects the mobility (fluidity) of the amphiphilic chains within the bilayer. In the gel phase, surfactant alkyl chains are fully extended, tightly packed, and well ordered, resulting in low membrane mobility. Upon heating above Tc, the chains adopt a more random, disordered arrangement, leading to increased fluidity in the liquid-crystalline phase [91,92].

The incorporation of Chol into the bilayer structure induces the formation of a third structural state, referred to as the liquid-ordered phase (Figure 4c). This phase exhibits intermediate characteristics, being less fluid than the liquid-crystalline phase yet less rigid than the gel phase [89]. Within the bilayer, Chol molecules are oriented such that their hydroxyl groups establish hydrogen bonds with the ester linkages of surfactants in proximity to the polar head groups, thereby reducing the mobility of adjacent hydrocarbon chains. At low Chol concentrations, this effect remains limited, and the liquid-ordered phase coexists with gel and liquid-crystalline phases. In contrast, at high concentrations, steric constraints imposed by Chol hinder the tight packing of fatty acid tails, resulting in the stabilization of the bilayer exclusively in the liquid-ordered state [89].

The bilayer thickness differs among the three phases due to variations in hydrocarbon chain orientation and packing density. It decreases markedly in the liquid-disordered phase (Lα) as a result of chain disorder and interpenetration, whereas in the liquid-ordered phase (Lo), Chol reduces chain orientation, leading to a thickness comparable to or slightly greater than that of the gel phase (Lβ) [93].

Several molecular features directly influence the Tc, including alkyl chain length, degree of unsaturation, charge, and head group type. The presence of double bonds within the hydrocarbon chains lowers Tc, thereby increasing chain fluidity and enhancing niosomal membrane permeability [34]. Conversely, an increase in chain length strengthens Van der Waals interactions, requiring greater energy to disrupt molecular packing and consequently elevating Tc [94]. Beyond its effect on permeability, Tc also governs bilayer stiffness, overall stability, APIs release kinetics and EE. Empirical studies have shown that surfactants with lower Tc values form vesicles with greater flexibility but reduced integrity, yielding more permeable, leak-prone NIOs compared to those composed of surfactants with longer alkyl chains [95,96].

The phase transition temperature of the surfactants used in niosomal formulations appears to influence the drug EE, with higher Tc values, typically associated with longer and saturated alkyl chains, generally leading to the formation of more rigid and less permeable bilayers, which can enhance APIs EE [33,97]. Within the Span series, Tc values are reported as 16–24 °C for Span 20, 42–47 °C for Span 40, 53–58 °C for Span 60, and –20 to –18 °C for Span 80, the latter showing markedly lower values due to the presence of an unsaturated double bond in its oleate side chain. While this trend suggests a correlation between Tc and EE, it is important to note that other formulation factors, such as Chol content, preparation method, and drug properties, also play significant roles.

#### 2.2.2. Additive Agents


*(a) Membrane Additives*


Chol, a steroid derivative, represents a key component of NIOs, as its concentration strongly influences properties such as EE and long-term stability. By intercalating within the surfactant bilayer, Chol forms hydrogen bonds between its hydroxyl group and the ester linkage near the polar head (Figure 5), thereby reducing chain flexibility, enhancing bilayer rigidity, and decreasing vesicle permeability, which limits APIs leakage [85]. Structurally, these interactions increase membrane heterogeneity and promote the formation of ordered lipid domains (lipid rafts), further contributing to mechanical stability and improved APIs EE [33].

The concentration of Chol required in NIOs is determined by the physicochemical properties of both the encapsulated drug and the surfactant, particularly in relation to the HLB value. Surfactants with higher HLB values (>6–10), characterized by bulky hydrophilic head groups, generally require higher Chol content (30–50 mol/L) to reduce the effective HLB of the formulation and ensure vesicle stability. Consequently, lauryl chain surfactants such as Brij 30 (HLB 9) and tetraglyceryl monolaurate (HLB 10.4) can form NIOs only in the presence of adequate Chol levels, whereas stearyl chain surfactants, including Brij 72 (HLB 4.9), glyceryl monostearate (HLB 3.8), and Span 60 (HLB 4.7), are capable of vesicle formation even in the absence of Chol [4,34].

The surfactant-to-lipid ratio is another critical determinant of EE. Increased surfactant-to-lipid ratio typically enhances drug encapsulation and system viscosity, up to an optimal point beyond which vesicle destabilization may occur [54]. Typically, NIOs formulations employ surfactant concentrations in the range of 10–30 mM (corresponding approximately to 1–2.5% *w*/*w*, depending on molecular weight [52,98]. Overall, Chol markedly contributes to the physical stability of NIOs by elevating the Tc, increasing vesicle hydrodynamic diameter, and ultimately enhancing EE [54,99].


*(b) Surface Additives*


Charge inducer molecules (ionic compounds) represent a distinct class of additives frequently employed in NIOs formulations. Their incorporation confers electrostatic charges to the bilayer, which exerts dual effects: (i) an expansion of the interlamellar distance in multilamellar vesicles, thereby increasing the total volume available for drug entrapment [45], and (ii) an enhancement of surface charge density, which reduces vesicle aggregation through electrostatic repulsion and improves overall stability [33]. Common examples include negatively charged molecules such as diacetyl phosphate (DCP) and phosphatidic acid, as well as positively charged molecules such as stearylamine (STR) and stearyl pyridinium chloride [85].

Charged NIOs offer several advantages, particularly for topical applications, where they can enhance skin permeability, improve APIs EE, and support the development of hybrid vesicular systems [4]. A notable example is the incorporation of DCP into gentamicin sulfate-loaded NIOs, which significantly increased API EE and ocular bioavailability compared to neutral vesicles [100].

The concentration of charge-inducing molecules is also critical. Optimal levels are reported to be within 2.5–5 mol/L, as higher concentrations may hinder vesicle formation [53]. Moreover, the extent of electrostatic stabilization is reflected in zeta potential values: complete stabilization occurs above 30 mV, limited flocculation is observed between 5 and 15 mV, while severe aggregation predominates when values fall between 3 and 5 mV [4].


*(c) Steric Additives*


Natural polymers such as polysaccharides and proteins, as well as synthetic polymers such as polyethylene glycol (PEG), are widely employed as steric additives to stabilize NIOs. By generating steric repulsion, these polymers reduce vesicle aggregation and fusion, thereby improving colloidal stability. Among them, PEG is most extensively used due to its excellent biocompatibility, solubility, and low toxicity. PEGylation further protects vesicles from enzymatic degradation and prolongs drug retention [86,101].

In addition, other polymers such as chitosan (CS), have been used as stabilizing agents. Unlike PEG, which stabilizes vesicles primarily through steric repulsion, CS exerts a combined stabilizing effect through both electrostatic and steric mechanisms. In systems containing negatively charged vesicle surfaces (typically achieved by incorporating charge-inducing agents), CS interacts via its positively charged amino groups with anionic groups at the bilayer surface, forming a thin polymeric layer. This electrostatic interaction is complemented by steric stabilization, hydrogen bonding with surfactant headgroups (hydroxyl or ether groups), and hydrophobic interactions with surfactant alkyl tails. Together, these interactions contribute to enhanced membrane rigidity, reduced bilayer permeability, increased surface charge density, and improved mucoadhesion and controlled drug release [102,103].

#### 2.2.3. Preparation Conditions

The preparation method employed in NIOs formulation critically influences vesicle size and lamellarity. Multilamellar vesicles (MLVs) or large unilamellar vesicles (LUVs) are typically obtained by thin-film hydration technique, whereas the reverse-phase evaporation method favors the formation of small unilamellar vesicles (SUVs). Microfluidization process is particularly advantageous, producing nanoscale vesicles with high uniformity [45]. Furthermore, regardless of the initial size distribution, MLVs can be converted into SUVs using conventional downsizing techniques such as sonication, high-pressure homogenization (microfluidization), or extrusion under high pressure [19]. Among preparation conditions, hydration play a decisive role in determining the physicochemical characteristics of NIOs.


*(a) Hydration Temperature*


The temperature of the hydration medium is a critical factor affecting vesicle formation, particularly size and morphology. Hydration must be conducted at temperatures above the Tc of the system to ensure successful bilayer formation [104]. When hydration occurs below Tc, the gel-to-liquid crystalline transition of the surfactant may not proceed adequately, resulting in defective vesicles [105].

Arunothayanun et al. [106] investigated polyhedral vesicles derived from hexadecyl diglycerol ether (C_16_G_2_) and poly-24-oxyethylene cholesteryl ether (Solulan C24), highlighting the influence of hydration temperature on vesicle morphology. Polyhedral NIOs formed from C_16_G_2_:Solulan C24 (91:9 molar ratio) exhibited characteristic thermal transitions, with a primary endothermic peak during heating and a main exothermic peak during cooling. At 25 °C, polyhedral vesicles predominated, which transformed into spherical vesicles upon heating to 48 °C. Cooling from 55 °C to 49 °C produced smaller spherical vesicles that reverted to polyhedral morphology at 35 °C. These findings indicate that vesicle ultrastructure is highly dependent on temperature-induced phase transitions, most likely related to changes in membrane flexibility. In contrast, spherical/tubular NIOs composed of C_16_G_2_:Chol:Solulan C24 (49:49:2 molar ratio) demonstrated no morphological transitions when heated from room temperature to 70 °C. This stability is attributed to Chol incorporation, which increases Tc and stabilizes the membrane in the liquid state.


*(b) Hydration Medium pH*


The pH of the hydration medium plays a critical role in determining EE of NIOs. Phosphate buffer at varying pH values is most commonly employed, with the selected pH depending on the solubility and stability profiles of the encapsulated APIs [54]. For instance, ascorbic acid-loaded NIOs have been prepared at pH 5 [107], while ketoconazole [108] and flurbiprofen [109] exhibited maximum entrapment at pH 5.5. At such acidic conditions, the unionized forms of these APIs predominate, favoring their partitioning into the lipid bilayer compared to the ionized species [54].


*(c) Hydration Medium Volume and Hydration Time*


The volume of the hydration medium and the duration of hydration are critical parameters in NIOs preparation, directly influencing vesicle characteristics and EE [98]. For example, methylene blue-loaded NIOs prepared by the thin-film hydration method exhibited larger vesicles with reduced EE when the hydration time was too short. Optimization studies established that a hydration time of 60 min and a hydration volume of 5 mL yielded stable NIOs with optimal EE [110]. Similarly, in zidovudine-loaded NIOs, EE improved as hydration time was extended from 20 to 45 min, whereas hydration volumes exceeding 6 mL promoted drug leakage [62].

#### 2.2.4. Characteristics of the Encapsulated APIs

The physicochemical properties of encapsulated drugs strongly influence NIOs bilayer flexibility, vesicle size, and surface charge. As vesicular carriers, NIOs can accommodate hydrophilic, lipophilic, or amphiphilic molecules, either individually or in associations. Entrapment within vesicles formed from NISs often increases particle size, most likely due to drug interactions with surfactant head groups, which enhance bilayer charge and repulsion [111]. This effect is mitigated in PEG-coated NIOs, where steric hindrance imposed by PEG chains restricts drug association with the bilayer, limiting particle growth [45]. In general, NIOs exhibit higher affinity for lipophilic APIs, which readily partition into the bilayer membrane, resulting in greater EE compared to hydrophilic drugs, which are confined to the aqueous core or adsorbed on the vesicle surface [111,112].

#### 2.2.5. Resistance to Osmotic Stress

Exposure of NIOs to hypo- or hyper-tonic environments, arising from physiological or pathological conditions, induces structural and functional alterations that compromise vesicle integrity. Such changes in bilayer organization affect both vesicle size and the release profile of encapsulated APIs [98,113]. In hypertonic solutions, osmotic gradients cause vesicle shrinkage, leading to the formation of “shrunken” NIOs. Conversely, in hypotonic media, vesicles initially swell with a gradual drug release, followed by accelerated release due to increased vesicle size and mechanical destabilization of the bilayer [45,98].

As summarized in Table 4, various classes of NISs exhibit distinct influences on NIOs characteristics such as EE, vesicle size, and stability, primarily determined by their molecular structure and physicochemical parameters (HLB, CPP, Tc).

### 2.3. Preparation Methods for NIOs

A variety of preparation methods have been reported for NIOs, selected according to the desired physicochemical and functional characteristics, including vesicle size, lamellarity, size distribution, EE, drug release behavior and intended therapeutic applications [39,52]. In all cases, the APIs to be encapsulated are introduced during the preformulation stage according to their solubility profile: lipophilic compounds are incorporated into the lipidic/organic phase together with NISs and Chol, while hydrophilic compounds are dissolved in the aqueous phase employed for hydration. The main techniques used for NIOs production, categorized according to vesicle size and lamellarity, are schematically illustrated in Figure 6.

#### 2.3.1. Thin-Film Hydration Method

The thin-film hydration (TFH) method, also referred to as the “hand-shaking method,” is one of the simplest and most widely used techniques for NIOs preparation [48]. Originally developed by Bangham et al. [114] for liposome fabrication, TFH has since been adapted for the preparation of various vesicular systems, including NIOs [115]. In this procedure, NISs, Chol, lipophilic additives are dissolved in an organic solvent within a round-bottom flask. The solvent is then removed under reduced pressure using a rotary evaporator, resulting in the formation of a thin lipid film on the inner surface of the flask [9]. After solvent evaporation, the resulting thin lipid film is hydrated either with an aqueous medium (if the API is already embedded in the lipid film) or with an aqueous drug solution containing hydrophilic APIs. Hydration is carried out under continuous agitation at a temperature above the Tc of the surfactant, leading to the formation of a milky niosomal suspension [116]. The TFH method typically yields MLVs with relatively large diameters. It has been successfully applied in the preparation of pentoxifylline-loaded NIOs for dermal delivery [117], as well as for doxorubicin [118] and curcumin- [119] loaded NIOs for tumor-targeted therapy.

#### 2.3.2. The Freeze–Thaw Method

The freeze–thaw method, a variation in TFH, is primarily used to produce MLVs. The procedure consists of repeated cycles of freezing the preformed niosomal suspension in liquid nitrogen, followed by thawing in a water bath at 60 °C [11]. This technique has been applied in the preparation of NIOs for ocular delivery of naltrexone hydrochloride [120] and for topical formulations of gallidermin [121].

#### 2.3.3. Transmembrane pH Gradient Method

The transmembrane pH gradient method, also known as the remote loading technique, is based on establishing a pH differential across the niosomal bilayer, whereby the intravesicular compartment is maintained under acidic conditions relative to the external medium. In this procedure, a thin lipid film is hydrated with acidic solution, typically citric acid at pH~4, to generate vesicles with an internal acidic environment [13]. After stabilization, commonly achieved through freeze–thaw cycling, the external medium is adjusted to near-neutral values (pH 7.0–7.2) using phosphate buffer. The subsequent addition of a weakly basic drug allows it to diffuse across the bilayer in its unionized form; once internalized, the drug becomes protonated within the acidic lumen and, consequently, is unable to diffuse back through the membrane. This “proton-trapping” mechanism enables efficient encapsulation of weakly basic APIs within the niosomal core [53].

#### 2.3.4. The “Bubble” Method

The “bubble” method represents a distinctive single-step process that eliminates the need for organic solvents but requires specialized apparatus, consisting of a three-neck round-bottom flask equipped with a water-cooled reflux condenser, a thermometer, and a nitrogen gas inlet [84]. In this approach, all components, phosphate-buffered saline (pH 7.4), NISs, additives, and APIs, are introduced simultaneously into the flask, which is maintained in a thermostatically controlled water bath [98]. To ensure successful vesicle formation, all constituents must remain in the liquid phase throughout the process. The dispersion is initially homogenized for approximately 15 s using a high-shear homogenizer, after which nitrogen gas is bubbled through the mixture at 70 °C, leading to the formation of LUVs. These vesicles can subsequently be downsized to obtain SUVs [104].

#### 2.3.5. Solvent Injection Method

In the solvent injection method, NISs, Chol, and other lipophilic additives are first dissolved in an organic solvent such as diethyl ether or ethanol. The resulting solution is then injected through a fine-gauge needle into a preheated aqueous phase maintained above 60 °C. Lipophilic APIs are solubilized in the organic phase, whereas hydrophilic APIs are introduced in the aqueous one prior to injection [84,122]. Gradual evaporation of the organic solvent leads to the formation of unilamellar vesicles (SUVs or LUVs) encapsulating the APIs. Owing to the use of flammable solvents, complete removal of residual organic solvent is required for safety and is typically achieved using rotary evaporation [52]. Applications of this technique include the encapsulation of hyaluronic acid for pulmonary delivery [123], pilocarpine hydrochloride for ocular administration [124] and stavudine for antiretroviral therapy [125].

#### 2.3.6. Sonication Method

This technique employs the dispersion of all the constituents, including Chol, NISs, APIs and other additives in a buffer solution that is further subjected to sonication at 60 °C for a certain time to produce SUVs [122]. The sonication method was used to prepare cefdinir [126] and diallyl disulfide [127] loaded NIOs.

#### 2.3.7. The Reverse-Phase Evaporation Method

The reverse-phase evaporation technique (REV) is based on the formation of a water-in-oil emulsion (W/O), in which the NISs, Chol, and other lipophilic additives are first dissolved in an organic solvent. An aqueous phase is then incorporated under continuous stirring to produce a biphasic system. Subsequent homogenization and sonication yield a stable emulsion, from which gradual removal of the organic solvent under reduced pressure at 40–60 °C promotes the transformation of inverted micelles into bilayered vesicular structures. In this method, hydrophilic APIs are directly dissolved in the aqueous phase, whereas lipophilic compounds are pre-associated with surfactants or co-dissolved in the organic phase as part of the lipid mixture. This process predominantly generates LUVs dispersed in the aqueous medium [128]. This approach has been successfully employed for the encapsulation of both low-molecular-weight compounds, such as clindamycin phosphate [129], salicylic acid [130], and fingolimod [131], as well as macromolecular agents such as bovine serum albumin [19].

#### 2.3.8. Microfluidization Method

Microfluidization represents a modern, high-precision technique for the preparation of SUVs, providing superior control over particle size, distribution, and reproducibility. The process is based on the high-pressure collision of two liquid streams, one aqueous and the other organic, within micro-engineered interaction chamber. The aqueous phase typically contains hydrophilic APIs, whereas the organic phase comprises NISs, Chol and other lipidic additives dissolved in a suitable organic solvent. At extremely high velocities, the streams impinge within the chamber, generating intense shear and turbulence forces that promote rapid mixing and spontaneous self-assembly of surfactant molecules into stable bilayered vesicles [132,133]. Comparative studies with conventional techniques such as TFH and solvent injection have demonstrated the advantages of microfluidization in producing Tween 80/Span 80-based NIOs with uniform size, narrow polydispersity, and enhanced reproducibility [37,133]. More recently, this method has been successfully employed for the encapsulation of balanocarpol [134] and curcumin [135], achieving improved encapsulation and enhanced therapeutic performance.

#### 2.3.9. The Heating Method

In this method, NISs and Chol are initially hydrated separately in phosphate buffer (pH 7.4) under a nitrogen atmosphere at ambient temperature to prevent oxidative degradation. Chol is dissolved by heating the medium to approximately 120 °C until complete melting, followed by gradual cooling to around 60 °C, at which point the surfactant solution and other excipients are introduced under continuous stirring. Lipophilic APIs can be incorporated during the Chol melting step, whereas hydrophilic APIs are added via the aqueous surfactant phase [115]. The resulting dispersion is maintained at room temperature for about 30 min to allow complete vesicle formation and subsequently stored at 4–5 °C under nitrogen to preserve structural integrity and prevent oxidation [11]. This technique has been successfully applied to prepare α-tocopherol-loaded NIOs exhibiting sustained drug release behavior [94].

#### 2.3.10. Proniosomes Technology

Proniosomes are advanced pro-vesicular carrier systems that exist in a dry or semisolid state that spontaneously form NIOs upon hydration with aqueous media [55]. Compared to conventional NIOs, they exhibit superior physicochemical stability, effectively overcoming problems related to aggregation, fusion, and leakage, while allowing on-demand preparation immediately before use. Lipophilic APIs are embedded within the dry lipidic matrix of proniosomes, whereas hydrophilic APIs are incorporated during the hydration step, when the system transitions into fully formed NIOs [136]. Depending on the preparation technique, proniosomes can be produced as free-flowing powders, typically through slurry or spray-coating methods, or as gels, commonly obtained through coacervation-phase separation [137]. Their versatility has been demonstrated in various drug-delivery applications, including tacrolimus [138], tazarotene [139], and roxithromycin [140].

Other Techniques:

In addition to the widely adopted preparation techniques, several alternative or derivative methods have been reported in the literature:*Lipid Injection Method*: a solvent-free modification of solvent injection technique, in which molten NISs, Chol, and additives are injected into a vigorously stirred and preheated aqueous phase, resulting in the formation of a niosomal suspension [9].*Emulsion Method*: involves the formation of an O/W emulsion by homogenizing an organic solution of NISs and Chol with an aqueous phase, followed by evaporation of the organic solvent to generate NIOs [45].*Dehydration-Rehydration Method*: originally described by Kirby and Gregoriadis [141], this approach produces MLVs by freeze-drying preformed NIOs obtained via TFH method, followed by rehydration with phosphate buffer (pH 7.4) at 60 °C [53].*Handjani-Vila Method*: involves mixing NISs and Chol with an aqueous phase under controlled temperature and agitation (or by ultracentrifugation), leading to the formation of a lamellar liquid-crystalline phase composed of homogeneous vesicles [104].*Multiple Membrane Extrusion*: a refinement of TFH method, in which hydrated niosomal dispersions are sequentially extruded through polycarbonate membranes (typically up to eight layers) to obtain vesicles with uniform and controlled size distribution [45,128].*Supercritical Carbon Dioxide (scCO*_2_*) Method*: developed by Manosroi et al. [142], this solvent-free, one-step technique produces LUVs by mixing formulation components with supercritical CO_2_ in a high-pressure glass cell under controlled temperature and pressure conditions (60 °C, 200 bar), eliminating the need for toxic organic solvents [115].*Enzymatic Method*: utilizes esterase-mediated cleavage of Chol esters or polyoxyethylene derivatives within mixed micellar systems, promoting the in situ formation of MLVs [33].*The Ball Milling Method*: a recent high-energy mechanical process in which APIs, NISs and Chol are co-processed in a rotating milling container containing grinding balls. The mechanical impact and shear forces induce particle fragmentation and compaction, leading to the formation of uniform NIOs [143,144,145].

Among the techniques described, several can be regarded as *conventional methods*, such as thin-film hydration, reverse-phase evaporation, solvent injection, or sonication, which were initially developed for liposomal systems and subsequently adapted for NIOs. These approaches remain widely employed due to their simplicity and reproducibility, being continuously optimized in recent publications [145]. In contrast, *modern and emerging methods* such as microfluidization, proniosome technology, supercritical CO_2_ method, and ball milling represent newer developments that offer improved scalability, solvent-free processing, and enhanced control over vesicle size and uniformity [144]. The coexistence of both classical and advanced approaches highlights the continuous evolution of NIOs fabrication toward more sustainable and industrially viable processes.

### 2.4. Purification and Characterization of NIOs

Ensuring the physicochemical stability and therapeutic performance of NIOs requires rigorous purification and comprehensive physicochemical characterization. Parameters such as vesicle morphology, size, bilayer architecture, zeta potential, size distribution, EE, stability, and in vitro release are critical quality attributes that determine formulation robustness, influencing both storage stability and in vivo performance. A summary of the principal purification strategies and characterization methodologies reported in the literature is provided in Table 5.

## 3. The Stimuli-Responsive NIOs—New Approaches for Targeting Drug Delivery

Beyond the intrinsic advantages of niosomal systems, recent advances have focused on the engineering stimuli-responsive NIOs that enable a more precise spatiotemporal control over drug release, thereby enhancing therapeutic efficacy [13,146]. Unlike conventional carriers, these smart nanovesicles are engineered to respond to specific endogenous or exogenous cues, ensuring site-specific delivery of the encapsulated agents [147,148].

Endogenous stimuli include various in pH, redox potential, or enzymatic activity, while exogenous stimuli comprise external factors such as light, ultrasound, magnetic fields, or electric trigger [149,150]. Temperature-sensitive carriers may belong to several categories, depending on whether the release is triggered by pathological processes (inflammation or tumor microenvironment) or by externally applied hyperthermia (ultrasound or high-frequency magnetic fields) [151].

Depending on the design, stimuli-responsive NIOs can exhibit reversible or irreversible behavior, and may be engineered as single-, dual-, or multi-responsive systems [152]. The release of the loaded APIs is initiated through structural transitions at the molecular or supramolecular level, such as protonation, hydrolytic cleavage, bond breakage, conformational changes, or degradation of surface layers, which act as programmed disintegration mechanisms [153]. Surface engineering further expands this concept by introducing functional moieties covalently or non-covalently attached to the vesicle. In such systems, drug release can be triggered by cleavage of the immobilized groups, effectively converting the carrier’s architecture into a responsive nanoplatform [154]. The mechanism by which drugs are released under the influence of internal or external stimuli in functionalized NIOS is represented in Figure 7.

### 3.1. pH-Responsive NIOs

pH-sensitive drug delivery systems are engineered to maintain the stability of the encapsulated APIs under physiological blood pH conditions (7.3–7.4), while enabling controlled and site-specific drug release within the acidic microenvironment characteristics of pathological tissues, such as malignant tumors, infected or ischemic regions, atherosclerotic plaques, and arthritic joints [155,156]. By exploiting local acidosis, these systems enhance therapeutic efficacy and minimize systemic toxicity, thereby overcoming the limitations of passive targeting associated with the enhanced permeation and retention [13,149].

Several design strategies have been employed to achieve pH-responsive NIOs. One of the most explored approaches involves the surface functionalization of vesicles with ionizable polymers bearing carboxylic acid or amino groups. Changes in environmental pH induce protonation or deprotonation of these groups, modifying the hydrophilic–hydrophobic balance of the nanocarrier and triggering conformational or solubility transitions that facilitate drug release [157].

An alternative strategy relies on the incorporation of acid-labile linkages such as hydrazone, ester, imine, oxime, or ketal bonds, which remain stable at neutral pH but undergo cleavage in acidic conditions, resulting in rapid and site-specific drug release [153,158]. Similarly, the inclusion of pH-responsive molecules such as cholesteryl hemisuccinate (CHEMS) can destabilize the bilayer structure under acidic conditions, promoting vesicle disassembly and facilitating drug release [13].

Finally, polymer coatings, such as PEG, hyaluronic acid, or folic acid, provide an additional level of control. When these polymers are linked via pH-cleavable bonds, they undergo detachment and degradation in acidic environments, thereby initiating drug release while simultaneously improving colloidal stability and prolonging systemic circulation prior to target site accumulation [151]. Table 6 summarizes the most relevant examples of pH-responsive niosomal formulations and the specific structural or surface modulations employed.

### 3.2. Temperature-Responsive NIOs

Thermo-responsive NIOs represent a distinct class of smart, stimuli-sensitive nanocarriers, extensively investigated in cancer therapy owing to the slightly elevated temperature of tumor tissues compared to normal counterparts [153]. These systems are designed to remain stable under physiological conditions (≤37 °C), preserving the encapsulated APIs, while enabling a controlled and temperature-triggered release when exposed to hyperthermic conditions (>40 °C). The release process can thus be activated either by externally applied thermal stimuli, such as radiofrequency, microwave, or high-intensity focused ultrasound, or by localized pathological hyperthermia occurring within diseased tissues [173].

Temperature sensitivity is typically conferred through the incorporation of thermo-responsive polymers, including poly[2-(2-methoxyethoxy)ethyl methacrylate], poly (amidoamine), poly(N-isopropylacrylamide), and poly(2-oxazoline)s, which undergo reversible phase or solubility transitions in aqueous media at their lower critical solution temperature (LCST). Such transitions induce structural rearrangements of the vesicle membrane, facilitating drug diffusion and release. Among the available materials, Pluronic^®^ L64, has been most frequently employed in the formulation of thermo-responsive NIOs. This amphiphilic block copolymer exhibits temperature- and concentration-dependent aggregation behavior, which directly influences vesicle stability and permeability [149]. Upon mild hyperthermia, Pluronic^®^ L64-coated NIOs demonstrate increased membrane fluidity and permeability, accelerating the diffusion of encapsulated APIs and achieving precisely controlled, temperature-triggered drug release [174,175]. Representative examples of thermo-responsive niosomal formulations are summarized in Table 7.

### 3.3. Magnetically Sensitive NIOs

Magnetic nanocarriers have attracted considerable interest in biomedical research due to their multifunctional properties, with applications spanning radionuclide therapy, magnetic resonance imaging (MRI) contrast enhancement, magnetic hyperthermia, and externally triggered release [151]. Their major key advantages include the capacity to be guided by an external magnetic field, enabling site-specific targeting, as well as the potential to biodegrade into physiological tolerable ions. Within drug delivery, magnetic NIOs offer prolonged systemic circulation, enhanced drug internalization into target tissues, reduced systemic clearance, and minimized off-target effects [151,179].

Formulation strategies typically involve the incorporation of magnetic nanoparticles, such as maghemite (γ-Fe_2_O_3_), magnetite (Fe_3_O_4_), hybrid composites (graphene/Au/Fe_3_O_4_), or zinc ferrite (ZnFe_2_O_4_), into the bilayer or core of the niosomal vesicles [149]. These systems are being extensively investigated in oncology, where magnetic targeting provides a means for spatiotemporal control of drug release and improved therapeutic efficacy [157,180]. Representative examples of magnetically responsive niosomal formulations, highlighting their composition, functional role of magnetic nanoparticles, and therapeutic implications, are summarized in Table 8.

### 3.4. Radio-NIOs

Radio-NIOs, or radio-labeled NIOs, represent a specialized subclass of nanocarriers designed for dual functionality, serving both therapeutic radionuclide delivery and diagnostic imaging purposes. These systems enable noninvasive monitoring of the pharmacokinetics and biodistribution of the encapsulated radionuclides through nuclear imaging techniques. Unlike other categories of stimuli-responsive NIOs, radio-NIOs act as carriers of the stimulus itself, the radionuclide, rather than responding to external or internal triggers [187].

Radiolabeling of vesicular nanocarriers can be accomplished via several radiochemical strategies, including: (i) direct coordination of the radionuclide to functional groups present on the vesicle surface, (ii) chelator-mediated complexation, in which the radionuclide binds to a lipid–chelator conjugate incorporated into the bilayer, and (iii) direct encapsulation of the radionuclide within the aqueous core during vesicle synthesis [188].

Among clinically employed the radionuclides, such as iodine-131[^131^I], gallium-67 [^67^Ga], and technetium [^99m^Tc], the latter remains the radionuclide of choice due to its favorable physical and chemical characteristics, including a 6 h half-life and emission of 140 keV γ-photons, which allow high-activity administration with limited radiation burden. Moreover, [^99m^Tc] exhibits optimal imaging properties for noninvasive single-photon emission computed tomography (SPECT) and gamma scintigraph, making it widely applicable in diagnostic and theranostic formulations [189]. Representative examples of radio-niosomal formulations, their labeling methods, and biodistribution outcomes are summarized in Table 9.

### 3.5. Multiple Stimuli-Responsive NIOs

In recent years, research on stimuli-responsive drug delivery systems has evolved toward the design of multifunctional nanocarriers capable of responding to multiple internal and/or external cues. These systems can be activated by combination of endogenous stimuli (e.g., pH–redox, pH–enzyme, pH–redox–enzyme) or by hybrid combination of endogenous and exogenous triggers (e.g., pH–light, pH–enzyme–light, magnetism–thermal–light) [193]. Such multifunctional architectures are engineered to overcome the limitations of single-stimulus nanocarriers, enabling more accurate spatiotemporal control over drug release and thereby enhancing therapeutic selectivity and efficacy within complex pathological environments.

In the context of cancer therapy, dual and triple-responsive systems are particularly promising, as tumor microenvironments are characterized by both acidic extracellular pH (~ 4.5–5.2) and locally elevated temperature (~ 42 °C), in contrast to the neutral pH (7.4) and normothermic conditions (37 °C) of healthy tissue [194]. Multiple-stimuli-responsive NIOs can thus be designed to exploit these pathophysiological differences, achieving synergistic activation, improved targeting and optimized therapeutic outcomes. Table 7 highlights representative multiple-stimuli-responsive niosomal formulations, emphasizing their structural modulations and therapeutic applications.

Another recently discovered variant of multistimuli-responsive niosomes is dual loading, as theranostic nanovesicular platforms that integrate both therapeutic and diagnostic functionalities. This is achieved when one of the encapsulated payloads acts as an imaging probe—such as quantum dots (CdSe/ZnS), iron oxide nanoparticles, near-infrared dyes (e.g., indocyanine green), gadolinium complexes, or Fe_3_O_4_ nanocrystals. Such hybrid systems enable simultaneous drug delivery and real-time imaging or monitoring of biodistribution. Such theranostic multiple-stimuli-responsive NIOs can be activated under internal (pH, enzyme, redox) or external (temperature, magnetic field, light) stimuli, allowing “on-demand” release of drugs and optical or magnetic signal generation for diagnostic tracking [195].

In cancer therapy, the formulation of theranostic NIOs has been widely applied to enhance treatment efficacy, offering several key advantages: (i) spatio-temporal control of drug release through stimuli-sensitive activation, (ii) minimization of systemic side effects by confining release to tumor tissue, (iii) real-time monitoring of biodistribution via optical or magnetic feedback from the imaging component, and (iv) the possibility of combined therapeutic modalities such as chemo–photothermal or chemo–imaging therapy [195,196]. This integrated strategy enhances treatment precision and provides clinicians with feedback on drug localization, accumulation, and therapeutic response. These multi-stimuli theranostic nanoplatforms exemplify the transition from conventional drug delivery to intelligent, image-guided nanomedicine.

The representative multiple-stimuli-responsive niosomal formulations, emphasizing their structural modulations and therapeutic applications are presented in Table 10.

### 3.6. Clinical Translation Status of Stimuli-Responsive NIOs and Conventional NIOs

Niosomes represent today one of the most promising nanotechnological platforms, proving in numerous experimental studies the ability to increase the stability of active substances, improve their solubility and implicit bioavailability, reduce systemic toxicity and moreover offer a sustained, prolonged release and a superior penetration of biological barriers. However, despite the significant progress made so far, no NIOs-approved therapy for any disorder has been approved by the Food and Drug Administration (FDA).

Stimuli-responsive NIOs, unlike conventional ones, represent a promising direction especially in anticancer drug delivery systems because it ensures a controlled and localized release of drugs, improved therapeutic targeting, increased anticancer efficacy, but nevertheless clinical translation is still in its infancy, with only a few preclinical experimental studies being conducted and with some examples presented in Table 11. The main negative factors influencing their clinical traceability are: limited scalability, reduced physicochemical stability (aggregation, fusion, and drug leakage over time), complex manufacturing processes which are reflected in high production costs but also reduced reproducibility from one batch to another, reduced stability during sterilization processes but also under certain storage conditions, as well as a potentially immunogenic character.

NIOs modified with anti-CD123 monoclonal antibodies were developed by Liu et al. [28] for targeted delivery of daunorubicin (DNR) to target tumor cells in acute myeloid leukemia. This innovative strategy of cytotoxic drug-monoclonal antibody conjugate incorporated into niosomal delivery system allowed for precise targeting of the drug to malignant cells, a significantly increased cellular uptake of up to 3.3-fold compared to unmodified NIOs demonstrated by flow cytometry, significantly enhancing the antitumor effect compared to standard niosomal formulations.

Bahrami-Banan and colleagues [197] developed an innovative thermos-responsive NIOs system for the controlled release of doxorubicin consisting of phosphatidylcholine (20%), Span 60 (52.5%), Chol (22.5%) and DSPE-PEG2000 (5%)—as a new personalized anticancer therapy. The results demonstrated that this niosomal formulation significantly enhanced the cytotoxic effect of doxorubicin on the leukemia cell line KG-1, confirming that thermos-responsive NIOs represent an efficient and versatile platform for improving therapeutic efficacy and reducing adverse effects associated with conventional chemotherapy.

The pH-sensitive copolymer-modified NIOs developed by Gugleva and colleagues [161] showed a higher Cur release rate in acidic media that can be extrapolated as a feasible approach in cancer therapy. In addition, they showed a higher inhibition rate of colony formation, T-24 cells, compared to the free drug. Thus, pH-sensitive NIOs can be explored as a feasible platform for curcumin targeting.

Regarding clinical studies conducted so far are very limited and include non-stimuli-responsive variants and most of them for local/topical uses. Several topical formulations of NIOs have reached clinical evaluation, such as NIOs carrying Triamcinolone acetonide for possible treatment of keloids, joint damage in rheumatoid arthritis and psoriatic arthritis, with proven superior histamine wheal suppression but with unknown clinical trial progress [198].

In a randomized clinical trial, topical application of niosomal zinc sulfate combined with intralesional Glucantime for cutaneous leishmaniasis demonstrated similar efficacy to standard protocol (cryotherapy plus Glucantime) suggesting it as a second-line therapy [200], but data from subsequent clinical trials are lacking.

A triple-blind randomized clinical trial shows that topical application of NIOs-zinc 2% in combination with cryotherapy significantly improved skin penetration, with significantly faster complete remission of lesions (up to 93.3%) associated with Verruca vulgaris disease, with minimal adverse effects and a reduced recurrence rate compared to conventional zinc formulation associated with cryotherapy but data from subsequent clinical trials are lacking [201].

A new niosomal formulation developed for oral tissue regeneration with topical application was the main subject of a randomized block placebo-controlled double-blind clinical trial [202]. In in vitro and clinical studies, the niosomal formulation for the release of anthocyanins (AC) demonstrated a controlled release, high physicochemical stability, superior permeability compared to conventional gel, support of fibroblast proliferation and migration, as well as the synthesis of collagen, fibronectin and laminin, significantly accelerating the healing process of oral wounds. Although there are no further clinical studies, these results confirm that NIOs are a promising platform for the efficient and safe delivery of bioactive compounds in tissue regeneration therapies.

Beyond these localized clinical applications, several clinical investigations have been conducted and completed, including the mucoadhesive film with NIOs as a carrier system for propolis designed to treat recurrent aphthous ulcers with clinical trial number NCT03615820 [199]. The clinical study demonstrated that treatment with mucoadhesive films with propolis-loaded NIOs resulted in a reduction in ulcer size (after 1–2 days) but also much faster healing compared to placebo.

Another niosomal formulation in advanced clinical trials, phase II/III with clinical trial number NCT05340725 [203], aims to evaluate the efficacy and safety profile of rectally administered dexmedetomidine niosomes in pediatric patients undergoing bone marrow aspiration for biopsy. These dexmedetomide NIOs may improve pain management through their sustained release properties of the active substance with analgesic and sedative properties, which would lead to longer-lasting pain relief. The results of this study have not yet been made public; they are under review by the National Library of Medicine. Three additional niosomal formulations have undergone clinical evaluation, although their results remain unpublished. The first involves NIOs carrying salbutamol sulfate, for which preliminary studies have demonstrated an extended release of up to 20 h compared with less than 2.5 h for the free drug. The second is a transmucosal niosomal gel containing melatonin (MLT), which showed a residence time of more than 3 h in ex vivo studies and, in a clinical study involving 14 healthy volunteers, improved the pharmacokinetic profile of MLT in a dose-proportional manner, enhancing absorption and increasing both Cmax and AUC while prolonging T1/2 [204]. The third clinical investigation assessed the effectiveness of transfollicular penetration of niosomal formulations used as carriers for minoxidil in the treatment of alopecia areata [205].

It can be concluded that the number of niosomal formulations translated into clinical trials remains limited, but nevertheless the fact that some have been initiated and completed indicates that there is translational interest in using niosomal formulations as drug delivery systems. Addressing this gap requires a comprehensive and interdisciplinary approach that includes the standardization of fabrication protocols, extensive safety and biodistribution assessments, and the development of customized regulatory frameworks specifically designed for smart nanocarriers [206]. Future progress will depend on simplifying responsive architectures and implementation of standardized analytical methods for assessing stimuli responsiveness, that factors are essential to enable regulatory approval and facilitating the clinical use of stimuli-responsive NIOs as advanced drug delivery systems.

## 4. Conclusions and Future Directions

This review has underscored the potential of NIOs as versatile vesicular drug delivery systems, highlighting their formulation components, preparation techniques, and functional modulation strategies. NIOs, primarily composed of NISs and Chol, exhibit several inherent advantages, including the capacity to encapsulate both hydrophobic and hydrophilic APIs, high biocompatibility and biodegradability, non-immunogenicity, and favorable storage stability. These characteristics collectively enable improved drug absorption, reduced systemic toxicity, and enhanced therapeutic efficacy. Beyond pharmacological applications, NIOs have also been explored as carriers for diagnostic agents, particularly in imaging, where they enhance contrast and diagnostic accuracy. Despite these advantages, several technical and translational challenges remain, such as vesicle aggregation, fusion, leakage of encapsulated agents, and difficulties in large-scale production, stabilization, and distribution. Addressing these limitations has been the focus of recent formulation engineering efforts, leading to the development of specialized and functionalized niosomal systems with improved colloidal stability, tunable release kinetics, and site-specific targeting capabilities. Such advancements have been achieved through both surface engineering (functionalization of hydrophilic head groups) and internal composition optimization (replacement of classical NISs with polymeric surfactants or integration of multifunctional payloads).

A major research trend has centered on the emergence of stimuli-responsive NIOs, capable of releasing their cargo selectively in response to endogenous or exogenous cues. These smart systems enable controlled, site-specific delivery, thereby minimizing systemic exposure while maximizing therapeutic outcomes. Considerable progress has been achieved in the development of pH-, thermo-, radio-, and magnetically responsive formulations, as well as dual- and multi-stimuli-sensitive carriers. Nevertheless, additional exploration is warranted to expand these approaches toward other triggering mechanisms, such as redox-, enzyme-, light-, or ultrasound-induced release, for which current evidence remains limited.

Looking forward, NIOs hold significant promise as next-generation nanocarriers for both therapeutic and theranostic purposes. The integration of modular designs principles with multi-stimuli responsiveness may enable the creation of personalized, adaptive and highly precise nanosystems. Ultimately, NIOs should be regarded not merely as passive carriers, but as versatile, modular platforms for smart nanomedicine, capable of integrating therapeutic agents, diagnostic probes, and stimuli-sensitive functionalities, thus paving the way for future advances in personalized therapy and clinical translation.

## Figures and Tables

**Figure 1 pharmaceutics-17-01473-f001:**
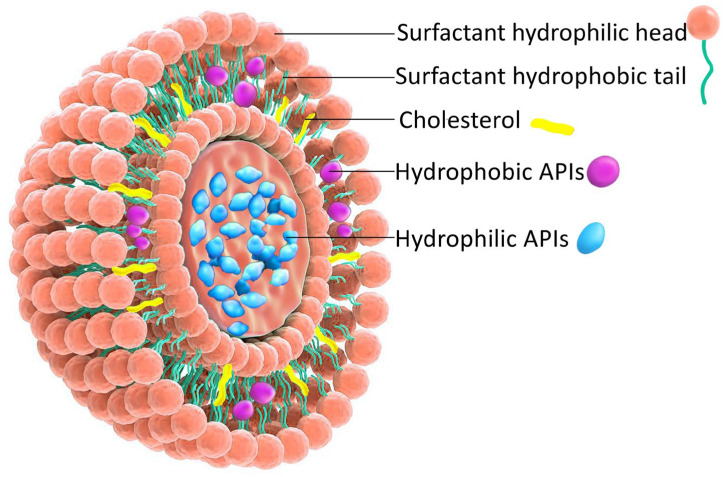
Schematic representation of the NIOs structure.

**Figure 2 pharmaceutics-17-01473-f002:**
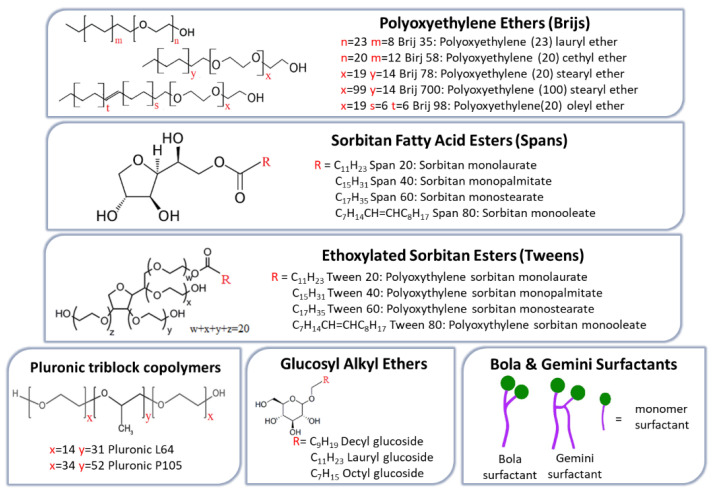
NISs commonly employed for NIOs preparation.

**Figure 3 pharmaceutics-17-01473-f003:**
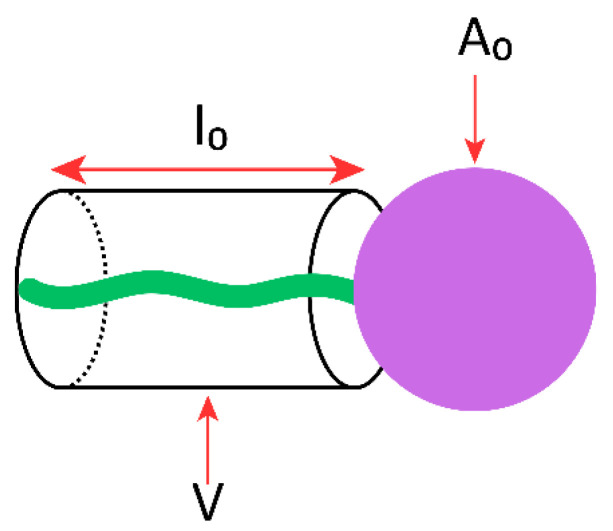
Schematic representation of a single-chain surfactant and its structural parameters used to define CPP: V, the hydrophobic chain volume; l_0_, the hydrophobic chain length; A_0_, the hydrophilic head group area.

**Figure 4 pharmaceutics-17-01473-f004:**
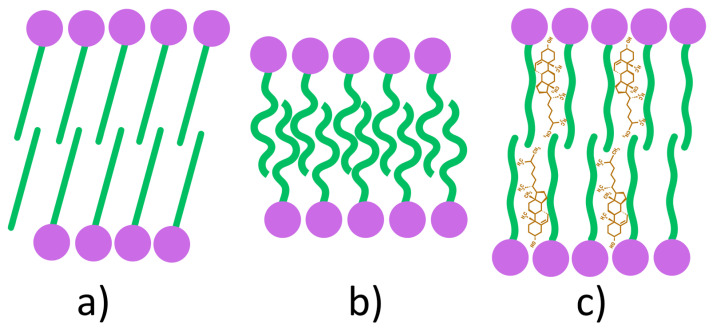
Schematic representation of phase transitions in amphiphilic bilayers: gel phase (**a**); liquid-disordered phase (**b**); liquid-ordered phase (**c**).

**Figure 5 pharmaceutics-17-01473-f005:**
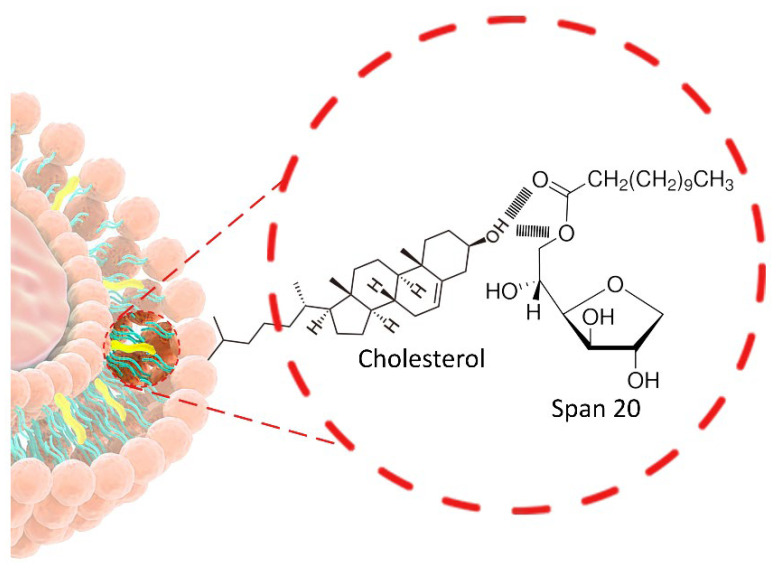
Structural localization of Chol within the surfactant bilayer.

**Figure 6 pharmaceutics-17-01473-f006:**
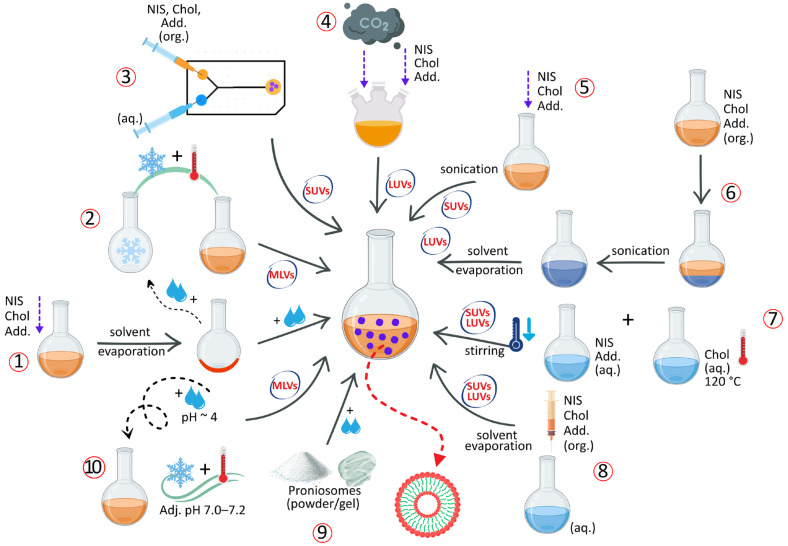
Schematic representation of NIOs preparation methods: thin-film hydration (1); freeze–thaw (2); microfluidization (3); “Bubble” method (4); sonication (5); reverse-phase evaporation (6); heating method (7); solvent injection (8); proniosomes technology (9); transmembrane pH gradient (10). Abbreviations: org. = organic phase; aq. = aqueous phase; NISs = non-ionic surfactants; Chol = cholesterol; Add. = additives; MLVs = multilamellar vesicles; SUVs = small multilamellar vesicles; LUVs = large multilamellar vesicles. Note: The incorporation of APIs follows their solubility profile, this step is not explicitly illustrated for the sake of clarity.

**Figure 7 pharmaceutics-17-01473-f007:**
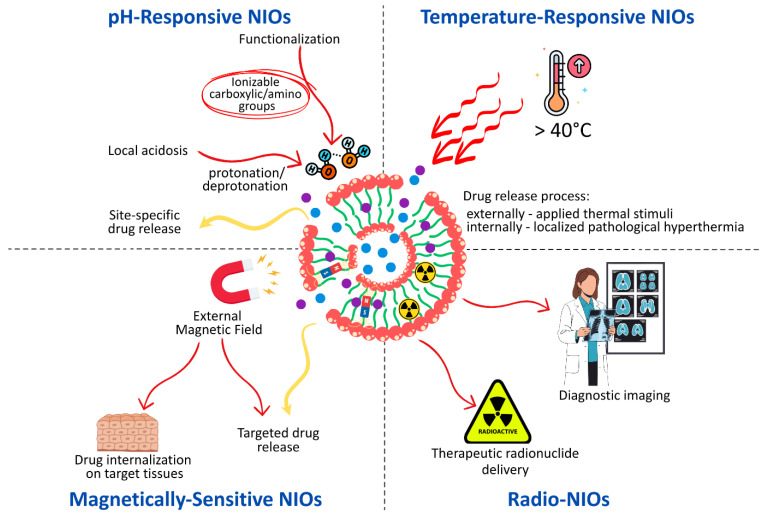
Mechanism of Stimuli-Induced drug release in Functionalized NIOSs.

**Table 1 pharmaceutics-17-01473-t001:** Classification of NIOs types.

Classification Criteria	Category	Representative Examples	Main Characteristics/Applications
Functionalized derivatives of NIOs	Ethosomes	Ethanol or isopropyl alcohol	Enhanced transdermal penetration
	Bola-surfactant NIOs	Bola surfactants (α, ω-hexadecyl-bis-(1-aza-18-crown-6))	Improved transdermal permeability
	Transfersomes	Edge activators	Ultra-flexible vesicles for membrane penetration
	Discomes	Cholesteryl poly-24-oxyethylene ether (Solulan C24)	Large disc-shaped vesicles for ocular drug delivery
	Aspasomes	Ascorbyl palmitate	Intrinsec antioxidant biological activity
	Elastic NIOs/Ethoniosomes	Ethanol or edge activators	Enhanced flexibility and deformability
	Polyhedral niosomes	Hexadecyl diglycerol ether (C_16_G_2_) or cholesteryl polyoxyethylene ether	Non-uniform spherical vesicles with a polygonal or faceted shape (typically 4–12 equal sides)
Functionalization purpose	Conventional NIOs	No special modifications	Conventional drug delivery
	Stealth NIOs (PEGylated)	PEG modification	Prolonged circulation, stealth behavior
	Targeted NIOs	Ligand-conjugated (antibody, peptide)	Targeted delivery to tumor/tissue
	Stimuli-responsive NIOs	Respond to pH, temperature, enzyme, or magnetic triggers	Controlled release and smart response
Lamellar structure	Unilamellar NIOs	~10–100 nm ~100–1000 nm	Small Unilamellar Vesicles (SUVs) Large Unilamellar Vesicles (LUVs)
	Multilamellar NIOs (MLVs)	~500–5000 nm	-
Encapsulated molecule type	Hydrophilic NIOs	APIs in aqueous core	Monotherapy
	Hydrophobic NIOs	APIs in bilayer	Monotherapy
	Co-loaded NIOs	Both hydrophilic and hydrophobic APIs	Combined therapies
Encapsulated agent	Theranostic NIOs	Drug & imaging agent	Dual therapy and real-time tracking
	Phytoniosomes	Plant-derived actives	
	Protein/Peptide-loaded NIOs	Proteins, enzymes or peptides	Targeted macromolecule delivery
	Gene-loaded NIOs	DNA, RNA, siRNA, miRNA	Gene therapy applications
	Hormone-loaded NIOs	Hormones (e.g., insulin, estradiol)	Controlled endocrine delivery
	Immunoniosomes	Antibody- or antigen-conjugated	Vaccine or immunotherapy systems
Preparation method	Conventional hydration	Thin-film hydration of surfactants + Chol	Common lab-scale preparation
	Proniosomes	Dried precursors in powder or gel	Enhanced stability, reconstitution in situ

**Table 2 pharmaceutics-17-01473-t002:** Advantages and disadvantages of NIOs.

Advantages	Disadvantages
High EE for both hydrophilic and hydrophobic drugs	Possible leakage of the entrapped drug
Biocompatible & biodegradable, being composed of non-toxic substances	Potential formation of niosomal aggregates during different preparation stages
Enhanced oral bioavailability and improved skin permeation	Risk of accumulation, fusion, or leakage of encapsulated drug in niosomal dispersions
Safe and non-toxic for administration via multiple delivery routes (oral, ocular, transdermal, parenteral etc.)	Variable encapsulation efficiency, especially for large hydrophilic molecules
Provide protection of encapsulated drugs against enzymatic degradation, oxidation, and other destabilizing processes	Potential local irritation depending on surfactants or excipients used
Effective carriers for targeted, controlled, and sustained drug delivery	Requirement of specialized equipment for certain preparation methods
	Challenges in scaling up from laboratory to industrial production

**Table 3 pharmaceutics-17-01473-t003:** Aggregation type of self-assembled amphiphiles correlated with CPP values.

CPP Value	Aggregation Shape	Surfactant Structural Characteristics	Schematic Representation
<1/2	Spherical or cylindrical micelles	Very bulky head (e.g., ethoxylate group) and fairly small tail length (*V* < *l_0_* × *A_0_* values)	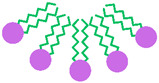
1/2–1	Geometrical packing (vesicles or flexible bilayers)	If *V*~*l_0_* × *A_0_* (CPP~1) the fairly symmetrical surfactant tends to be packed into cubic or simple lamellar liquid crystalline (Lα) phases, which when dispersed into water form vesicles	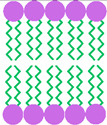
>1	Inverted micelles	Cone-shaped surfactant with a very bulky tail and a small head and/or short tail (*V* > *l_0_ *× *A_0_* values)	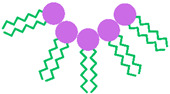

**Table 4 pharmaceutics-17-01473-t004:** Summary of formulation factors and materials influencing the characteristics of NIOs.

Formulation Factor	Key Material	Observed Effect on NIOs	Ref.
*NISs*			
Alkyl ether surfactants	Alkyl glyceryl ethers; Brij series;	Stable, low-allergenic vesicles; suitable for macromolecule delivery.	[31,55,56,57]
Alkyl ester derivatives	Span series	Non-toxic, non-irritant; EE ↑ with chain length and saturation (Span 60 > Span 40 > Span 20 > Span 80); longer saturated chains ↑ bilayer rigidity and stability.	[58,59,60,61]
Alkyl ester derivatives	Tween series	Suitable for hydrophilic drug encapsulation; EE ↓ with chain length (Tween 20 > Tween 60 > Tween 40 > Tween 80); Tween 80 → gene delivery; Tween 20 → epithelial permeability	[58,62,63]
Pluronic triblock copolymers	EO–PO–EO type (poloxamers)	Improve EE and stability; suitable for injectable and thermo-responsive NIOs; Poloxamer 184 ↓ Chol need in mixed systems.	[64,65,66,67,68]
Glucosyl alkyl ethers (glucosides & alkyl polyglucosides)	Myristyl-, cetyl-, stearyl-glucosides; Octyl-/decyl-polyglucosides	Biodegradable, non-toxic; longer chains → stable vesicles; enhance transdermal and cutaneous drug delivery (e.g., tretinoin).	[69,70,71]
Bola & Gemini surfactants	Bola (α,ω-type) & Gemini dimers	Bola surfactants: ↑ water solubility, ↑ CMC, ↓ aggregation; Gemini surfactants: ↓ CMC, ↑ micelle stability and solubilization; both → non-toxic and non-hemolytic.	[72,73,74,75,76,77]
Fatty alcohols & fatty acids	Stearyl-, cetyl-, & myristyl-alcohol; Stearic-, palmitic-, myristic-, oleic-, linoleic-, octanoic-, & decanoic-acid	Form bilayer vesicles affecting size, stability, and EE; fatty alcohols → SUVs with controlled release; fatty acids → larger vesicles, stability pH-dependent (near pKa); mixtures improve bilayer stability.	[78,79,80]
*Key NIS characteristics*			
Hydrophilic–Lipophilic Balance	Span, Tween, Brij series	HLB 4–8 → stable vesicles, ↑ EE (Span 60, Brij 72); high HLB → larger vesicles, ↓ EE (Tween 80, Brij 76); Chol needed for bilayer stabilization.	[59,60,61,81,84,85]
Critical Packing Parameter	-	Influences aggregation and bilayer assembly: ~ 0.5–1 → vesicles; <0.5 → micelles; >1 → inverted micelles.	[86,87,88]
Gel–Liquid Transition Temperature	-	Higher Tc → rigid, stable vesicles, ↑ EE; Lower Tc → flexible, permeable bilayers; Chol stabilizes bilayer via liquid-ordered phase.	[89,94,95,96,97]
*Additive Agents*			
Membrane Additives	Chol	Intercalated in bilayer → ↑ rigidity, Tc, and EE, ↓ permeability; high-HLB surfactants need 30–50 mol % Chol; optimal surfactant/lipid ratio (10–30 mM) improves stability.	[33,34,52,54,98,99]
Surface additives (charge-inducing agents)	(–): DCP, phosphatidic acid; (+): stearylamine, stearyl pyridinium chloride	Provide electrostatic stabilization; ↑ EE, ↓ aggregation; optimal 2.5–5 mol%.	[4,33,45,85]
Steric Additives	PEG, CS	Improve colloidal stability via steric (PEG) or combined steric–electrostatic (CS) mechanisms; ↓ aggregation, ↑ rigidity, ↑ mucoadhesion, and prolong drug release.	[101,102,103]
*Preparation Conditions*			
Hydration Temperature	-	Above Tc → stable bilayer formation; below Tc → defective vesicles; can effects vesicle morphology (polyhedral ↔ spherical); Chol improves thermal stability.	[105,106]
Hydration Medium pH	Phosphate buffer with various pH values	Acidic pH (5–5.5) enhances EE by favoring unionized APIs forms and better bilayer incorporation.	[54,107,108,109]
Hydration medium volume & time	-	Optimal hydration parameters → ↑ EE, stable vesicles; Short time or excess volume → large vesicles, drug leakage.	[62,98,110]
*Other Formulation-Dependent Factors*			
Characteristics of the Encapsulated APIs	Hydrophilic vs. lipophilic drugs	Lipophilic APIs → ↑ EE (bilayer partitioning); Hydrophilic → ↓ EE (aqueous core); PEGylation limits size increase via steric hindrance.	[45,111,112]
Resistance to osmotic stress	Hypo-/hypertonic media	Hypertonic → vesicle shrinkage; hypotonic → swelling and accelerated drug release due to bilayer destabilization.	[45,98,113]

Abbreviations: APIs, Active Pharmaceutical Ingredients; Chol, cholesterol; CMC, critical micelle concentration; CS, chitosan; DCP, diacetyl phosphate; EE, entrapment efficiency; EO, ethylene oxide; HLB, hydrophilic–lipophilic balance; NISs, non-ionic surfactants; NIOs, niosomes; PEG, polyethylene glycol; PO, propylene oxide; SUVs, small unilamellar vesicles; Tc, gel-liquid crystalline transition temperature; ↑—increase, ↓—decrease

**Table 5 pharmaceutics-17-01473-t005:** Purification and physicochemical characterization of NIOs.

Specific Properties	Methodology	Reference
Purification		
	Dialysis of the aqueous niosomal dispersion against phosphate buffer, normal saline, or glucose solution	[4,9,33,83,86]
	Gel filtration chromatography (Sephadex) with elution in phosphate buffer or normal saline	
	Centrifugation/ultracentrifugation of the niosomal suspension, followed by removal of the supernatant and resuspension of the pellet	
Characterization		
Vesicle size and surface morphology	Dynamic light scattering (DLS); scanning electron microscopy (SEM); scanning tunneling microscopy (STM); transmission electron microscopy (TEM); cryo-TEM; atomic force microscopy (AFM); freeze fracture replication-electron microscopy (FF-TEM); negative-stained transmission electron microscopy (NS-TEM)	[11,13,53,85,112]
Size distribution and polydispersity index	Dynamic light scattering (DLS)	[13,34,53,86]
Charge of vesicle and zeta potential	Dynamic light scattering (DLS); microelectrophoresis; pH-sensitive fluorophores; high-performance capillary electrophoresis	[53,55,85,86,115,122]
Bilayer characterization	Lamellarity: AFM, nuclear magnetic resonance (NMR); small angle X-ray scattering (SAXS) Membrane rigidity: fluorescence polarization with DPH (1,6 diphenyl- 1,3,5-hexatriene) Bilayer thickness: fluorescence polarization combined with the in situ energy-dispersive X-ray diffraction (EDXD)	[11,13,44,85,86,115]
Vesicle stability	Evaluation of mean vesicle size, size distribution, and EE% during storage at various temperatures, combined with photodegradation studies (UV irradiation and fluorescent light exposure)	[33,34,54,86]
Entrapment efficiency	EE is determined by separating the unencapsulated (free) APIs from the vesicular fraction, followed by quantification of the encapsulated APIs content using appropriate spectrophotometric or chromatographic methods	[11,55,85,86,115,122]
In vitro drug release	The drug release profile is evaluated using the dialysis method, in which niosomal dispersions are placed within semipermeable membranes and subjected to controlled experimental conditions. Samples are periodically withdrawn from the external dialysate at predefined intervals, and the drug content is quantified using validated spectrophotometric or chromatographic analytical methods	[11,13,85,86,115]
In vivo studies	Drug biodistribution, tissue accumulation (in organs such as the liver, lungs, spleen, and bone marrow), and residence time are assessed in animal models following administration, depending on the route of delivery and the administered drug concentration	[53,55,98]

**Table 6 pharmaceutics-17-01473-t006:** Representative pH-sensitive niosomal formulations.

Specific Modulations	Biological Model and Therapeutic Agent	NIOs Composition and Preparation	Remarks and Applications	Ref.
CS-based pH-sensitive NIOs	In vitro model using doxycycline	Span 80/Tween 80; supercritical carbon dioxide-assisted process	CS coating decreases gastric release at pH 1.2 and enabled controlled intestinal/colonic delivery; diffusion mechanism remained Fickian, with coating layer acting as protective layer without altering release kinetics	[159]
Tween-based pH-sensitive NIOs	Human ovarian cancer cell lines (OVCAR-3, OVSAHO, Kuramochi) treated with all-trans retinoic acid (ATRA)	Tween (20, 21)/Chol; TFH method	NIOs loaded with ATRA showed stable encapsulation and serum compatibility; demonstrated pH-dependent release (slow at 7.4, fast at 5.5) and improved efficacy in HGSOC models	[160]
HD-PAA_12/17_-based pH-sensitive NIOs	Human tumor cell lines, urinary bladder carcinoma (T-24), cutaneous T-cell lymphoma (HUT-78, MJ), and lung carcinoma (H1299) and L929 murine fibroblasts (control) treated with curcumin and calcein (model marker)	Span 60/Tween 60/Chol/HD-PAA copolymers; TFH method	HD-PAA modification enabled pH-responsive drug release (in acidic media), enhanced internalization, and strong pro-apoptotic effect; most pronounced in T-24 bladder carcinoma cells	[161]
PEGylated surface-modified NIOs	AGS human gastric adenocarcinoma cell line (C131) treated with cyclophosphamide; HFF fibroblasts (C163) used as control	Span 20/Span 60/Chol/PEG 2000; TFH method	pH-responsive release under tumor-like conditions; enhanced cytotoxicity and apoptosis compared to free cyclophosphamide and non-PEG NIOs; selective G2/M cell cycle arrest, halting cell division at the G2 (Gap 2) to M (mitosis) transition, a key mechanism commonly triggered by cytotoxic agents	[162]
HA/FA/PEG-based surface-functionalized NIOs	MCF-7 (human breast cancer), 4T1 (mouse breast cancer) treated with 5-fluorouracil; MCF-10A (non-tumorigenic epithelial cells) used as control	Span 60/Chol; TFH method	Surface-functionalized NIOs enhanced tumor cell uptake and apoptosis; FA-decorated NIOs showed highest cytotoxicity, reduced migration, and strongest cellular uptake in acidic pH (5.4); sustained release at pH 7.4; ROS elevation and necrosis/apoptosis confirmed via flow cytometry	[163]
CHEMS-based pH-sensitive NIOs	Breast cancer cells treated with carfilzomib (CFZ)	Span 60/Tween 60/ergosterol/ CHEMS; TFH method	pH-dependent release (74.39% at pH 5.4 vs. 54.55% at pH 7.4); enhanced cytotoxicity in breast cancer cells (IC_50_ = 0.0415 µM vs. 0.0714 µM for free CFZ); synergistic effect with doxorubicin (combination index <1); safe ≤ 2 mg/kg, toxicity at 4 mg/kg	[164]
	HCC: HepG2, Huh7 tumor cells (in vitro); AKT/c-Met–induced HCC mouse model (in vivo); treated with Tanshinone IIA	Span 80/Chol/CHEMS; ethanol injection method	Functionalization with galactose ligands enhanced selective uptake in hepatic tumors; pH-sensitive release increased antitumor efficacy of Tanshinone IIA; significant apoptosis induction (Annexin V/PI staining); G0/G1 cell cycle arrest; improved pharmacokinetics and biodistribution (higher plasma and liver retention); superior tumor suppression in AKT/c-Met–induced HCC mice	[165]
CHEMS-based pH-sensitive NIOs	MCF7 breast cancer cells treated with methotrexate (MTX); HUVECs (control); Wistar rats (in vivo toxicity)	Tween 60/Span 60/ergosterol/ CHEMS; TFH method	pH-dependent release (initial burst release, 42.1% in 4 h, followed by a slower release phase up to 24 h, reaching 79.4% at pH 5.4, vs. a sustained release of 51.2% after 24 h at pH 7.4); lower IC50 in MCF7 cells for pH-MTX/NIOs vs. free MTX (9.46 vs. 84.03 μg/mL); reduced toxicity in normal cells and in vivo (2 mg/kg safe; 4 mg/kg induced liver/kidney damage)	[166]
	MCF7 human breast cancer cells treated with cisplatin	Span 60/Tween 60/ergosterol/ CHEMS; TFH method	Sustained and pH-responsive release (faster at pH 5.4 vs. delayed at pH 7.4); enhanced cytotoxicity vs. free cisplatin; improved cell-killing effect; high EE% (89%)	[167]
	MCF-7 (breast cancer), OVCAR-3 (ovarian cancer) treated with mitoxantrone; HUVECs used as control	CHEMS/ PEG-PMMI-CholC6 copolymer; modified ethanol injection method	PEG-PMMI-CholC6 copolymer conferred pH-sensitivity and plasma stability; enhanced mitoxantrone release in acidic conditions (pH 6.5); higher cytotoxicity in tumor cells vs. conventional NIOs; lower toxicity in HUVECs	[168]
	MCF7 (human breast cancer), HeLa (human cervical cancer) treated with paclitaxel; HUVECs used as control	Span 60/Tween 60/ergosterol/ CHEMS; TFH method	pH-responsive release (faster at pH 5.2 vs. 7.4); EE% of 77%; improved antitumor efficacy (breast, cervical); lower IC_50_ vs. free PTX; reduced toxicity at 2.5 mg/kg dose; safer profile in normal cells	[156]
Calcium alginate-coated pH-sensitive NIOs	MDA-MB-231 and SKBR3 (breast cancer cells) treated with curcumin; MCF-10A (normal breast epithelial cells) used as control	Span 80/Chol/calcium alginate coating; TFH method	pH-dependent release: sustained at pH 7.4, accelerated at pH 3; enhanced apoptosis and improved selectivity toward cancer cells; biocompatibility in normal cells	[169]
pH-sensitive Tween 20 derivatives with modified polar head groups	CD-1 mice with chemically induced inflammatory and neuropathic pain, treated with ibuprofen	Tween 20 derivatives (GLY)/Chol; TFH method	Bilayer destabilization at low pH enabling site- specific ibuprofen release; antinociceptive effects in multiple mouse pain models: writhing, capsaicin, zymosan-induced hyperalgesia, and CCI-induced allodynia; superior performance vs. free ibuprofen	[155]
pH LIP-based surface-functionalized NIOs	A549 lung carcinoma, 4T1 mammary carcinoma cells; tumor tissue (mouse model); R18 (fluorescent membrane tracer)	Span 20/Chol - pHLIP conjugates (DSPE-pHLIP or Pyr-pHLIP); TFH method	pH-sensitive targeting enabled selective accumulation in tumors with 2–3 fold increased uptake and prolonged circulation vs. PEG-NIOs; confirmed pH-dependent uptake and minimal toxicity; effective tumor imaging and biodistribution	[170]
pH-sensitive Tween 20 derivatives with modified polar head groups	Inflamed tissues in CD-1 mice treated with ibuprofen or lidocaine; in vitro cytotoxicity assessed on HaCaT (human keratinocytes) and Balb/3T3 (mouse fibroblasts)	Tween 20 or Tween 20 (GLY)/Chol; TFH method	Improved pH-responsive release via bilayer destabilization by protonation of glycine residues; non-cytotoxic; modulated in vivo antinociceptive and anti-inflammatory activity (formalin and zymosan paw edema models); higher bilayer fluidity vs. non-pH-sensitive NIOs	[171]
	Hepatoblastoma cells; calcein (model marker)	Tween 20 derivatives (GLY, MMG, DMG)/Chol; TFH method	Proton sponge effect of Tween-20 derivatives enabled pH-sensitive intracellular delivery and enhanced calcein release; efficient uptake in 15 min; low toxicity (>93% viability)	[172]

Abbreviations: CCI, chronic constriction injury (neuropathic pain model); CHEMS, cholesteryl hemisuccinate; Chol, cholesterol; DMG, N,N-dimethyl-glycine; DSPE, 1,2-dioleoyl-sn-glycero-3-phosphoethanolamine (lipid anchor); FA, folic acid; GLY, glycine; HA, hyaluronic acid; HaCaT, human adult low calcium high temperature keratinocytes; HCC, hepatocellular carcinoma; HD-PAA, hexadecyl-poly(acrylic acid) polymers; HFF, human foreskin fibroblasts; HGSOC, high-grade serous ovarian cancer; HUVEC, Human Umbilical Vein Endothelial Cell (non-cancerous human endothelial cells used as control); MMG, N-methyl-glycine; MTX, methotrexate; TFH, thin-film hydration.

**Table 7 pharmaceutics-17-01473-t007:** Representative thermo-responsive niosomal formulations.

Specific Modulations	Biological Model and Therapeutic Agent	NIOs Composition and Preparation	Remarks and Applications	Ref.
Fatty acid-based thermo-responsive NIOs	Gram-positive (*Bacillus subtilis*, *Staphylococcus epidermidis*, *Staphylococcus aureus*, *Listeria monocytogenes*, *Enterococcus faecalis*) and Gram-negative (*Escherichia coli*, *Acinetobacter baumannii*, *Klebsiella pneumoniae*) bacteria exposed to tetracycline	Span 60/Chol/PCM (lauric & stearic acids eutectic mixture); TFH method	Temperature-triggered release system with low leakage at 37 °C and enhanced release at 42 °C; improved antibacterial efficacy (MIC for *S. epidermidis* reduced from 112.81 to 14.10 μM at 42 °C); promising candidate for heat-augmented infection therapy	[176]
FA-functionalized thermo-sensitive NIOs	Human breast cancer cells (MCF-7, MDA-MB-231, SKBR-3) and normal breast epithelial cells (MCF-10A) treated with doxorubicin and/or CQDs (photothermal & bioimaging agents)	Tween 60/Span 40/Chol/FA/ lecithin; TFH method	Thermo-sensitive folate-targeted NIOs with CQDs: lecithin enhanced membrane fluidity and enabled thermally triggered doxorubicin release >41 °C; effective chemo-photothermal cytotoxicity in cancer cells; selective uptake via folate receptors	[177]
Pluronic^®^ L64-based thermo-responsive NIOs	In vitro release study: calcein (model marker), 5-Fluorouracil (API)	Pluronic^®^ L64 (or L64ox)/Chol; TFH method	Thermo-sensitive release enhanced at 42 °C vs. 25–27 °C; temperature-dependent release attributed to surfactant structure, not drug type; promising stealth and targeted delivery	[174]
	Physicochemical model (1-Naphthol, fluorescent model)	Span 60/Pluronic^®^ L64/Chol; TFH method	Tunable phase transition (Tc ~ 40 °C); fluorescence & Raman analysis confirmed bilayer transition; suitable for thermo-responsive drug delivery under mild hyperthermia	[178]

Abbreviations: Chol, cholesterol; CQDs, carbon quantum dots; FA, folic acid; NIOs, niosomes; NIR, near-infrared Radiation; PCM, phase change material; L64, Pluronic^®^ L64; L64ox, oxidized derivative of Pluronic^®^ L64; MIC, Minimum Inhibitory Concentration; TFH, thin-film hydration.

**Table 8 pharmaceutics-17-01473-t008:** Representative magnetically sensitive niosomal formulations.

Specific Modulations	Biological Model and Therapeutic Agent	NIOs Composition and Preparation	Remarks and Applications	Ref.
Fe_3_O_4_-based magnetically responsive NIOs	In vitro release study; doxorubicin as model drug	Span 60/Chol/Fe_3_O_4_; TFH method	AMF-triggered burst release (86% in 3 h vs. 3% in 30 days); 1st order kinetics; magnetic loading conferred precise AMF-responsive control and enhanced efficiency	[181]
	Human embryonic kidney cell line (HEK-293T) exposed to plasmid DNA/protamine complex	Span 60/Tween 60/Chol or ergosterol/Fe_3_O_4_; TFH method	Enhanced uptake, magnetic responsiveness, and gene expression in HEK-293T cells; ergosterol-based vesicles showed smaller size, better stability, slower plasmid release, and higher transfection efficiency vs. Chol-based ones; magnetic nanoparticles facilitated guided delivery and enhanced performance under magnetic stimulus	[182]
	Human chronic myelogenous leukemia cell (K562) treated with doxorubicin	Tween 60 or Pluronic L64/Fe_3_O_4_; TFH method	Magnetically triggered release of doxorubicin via co-encapsulation of Fe_3_O_4_ nanoparticles; sustained drug release (50% in 5 h vs. 100% in 3 h for free drug); low inherent cytotoxicity of carriers supports tumor-targeted applications	[180]
PEGylated magnetically responsive NIOs	A549 lung cancer cells treated with metformin (MET) and artemisinin (ART); HBE normal epithelial cells used as control	Span 60/Chol/Fe_3_O_4_/ PEG 4000; TFH method	PEGylated Fe_3_O_4_-loaded NIOs co-delivering ART and MET: synergistic cytotoxicity, enhanced uptake and apoptosis in A549 cells; magnetic field (1.3 T) improved drug targeting and reduced viability (from 60% to 45%); no significant cytotoxicity in HBE cells	[183]
	Breast cancer cells (BT-474) treated with LFG-specific siRNA and chemotherapeutics (erlotinib/trastuzumab)	Span 60/Chol/Fe_3_O_4_/DSPE-PEG; TFH method	PEG-maleimide-functionalized Fe_3_O_4_-NIOs enhanced siRNA protection, intracellular uptake, and gene silencing; induced apoptosis and amplified drug efficacy (erlotinib/trastuzumab); magnetic field further boosted uptake	[184]
	Breast cancer cells (MCF-7) treated with AuNPs (SERS probe)	Span 60/Chol/Fe_3_O_4_/ DSPE-PEG; TFH method	Tf-functionalized Fe_3_O_4_/AuNP-loaded NIOs enabled dual targeting (magnetic and Tf-mediated) and SERS-based sensing; high intracellular accumulation in MCF-7 cells; PEGylated niosomal shell provided enhanced structural stability and contamination-free performance, supporting accurate tumor monitoring	[185]
	Breast cancer cells (MCF-7) treated with carboplatin	Tween 60/Span 60/ Chol/Fe_3_O_4_@SiO_2_/PEG 6000; TFH method	PEGylated Fe_3_O_4_@SiO_2_ NIOs enabled magnetic targeting and sustained carboplatin release; enhanced cytotoxicity observed in MCF-7 cells with external magnetic field (38% vs. 57%); PEGylation reduced premature release and improved bioavailability	[186]

Abbreviations: AMF, alternating magnetic field; AuNPs, gold nanoparticles; Chol, cholesterol; PEG, polyethylene glycol, siRNA, small interfering RNA; DSPE-PEG, 1,2-Distearoyl-sn-glycero-3-phosphoethanolamine–polyethylene glycol; HBE, Human Bronchial Epithelial cells (non-cancerous cell line); LFG, lifeguard; NIOs, niosomes; SERS, Surface-Enhanced Raman Scattering; Tf, transferrin; TFH, thin-film hydration.

**Table 9 pharmaceutics-17-01473-t009:** Representative radio-niosomal formulations.

Specific Modulations	Biological Model and Therapeutic/Diagnostic Payload	NIOs Composition and Preparation	Radiolabeling Strategy	Remarks and Applications	Ref.
Directly radiolabeled NIOs	Human colorectal adenocarcinoma (HT-29) cells exposed to [^99m^Tc]-radiolabeled NIOs	Span 60/Tween 60/Chol; TFH method	Direct [^99m^Tc] labeling via SnCl_2_ reduction; reduced Tc coordinates with functional groups of the NIO bilayer	High radiolabeling efficiency (>95%), stability up to 6 h in biological media, and enhanced uptake in HT-29 cells versus free [^99m^Tc]; suitable for passive targeting via EPR effect; outperforming chelator-based methods	[188]
	Ehrlich ascites carcinoma cells (solid tumors induced in mice) exposed to intravenously administered [^131^I]-ACM-loaded NIOs	Span 60/Chol; ether injection method	Direct [^131^I] radiolabeling of ACM via electrophilic substitution using chloramine-T	Enables dual chemo-radiotherapeutic functionality: ACM mediates PGE_2_-dependent chemotherapy, while [^131^I] delivers tumor-targeted radiotherapy with enhanced uptake, tumor regression, metastasis suppression, and tissue-specific accumulation; supports passive targeting via the EPR effect	[190]
Chelator-mediated radiolabeled PEGylated NIOs	Mouse breast tumor cells (4T1), exposed to [^99m^Tc]-radiolabeled PEGylated NIOs for passive tumor targeting	Span 60/Chol/TPGS-DTPA; Sonication method	Surface chelation of [^99m^Tc] using TPGS-DTPA in the presence of SnCl_2_ as a reducing agent	Stable [^99m^Tc]-NIOs with high radiolabeling efficiency (98%) and in vitro/in vivo stability; tumor accumulation confirmed via scintigraphic imaging and biodistribution; high tumor-to-muscle uptake ratio via EPR; validated prototype for passive tumor targeting	[187,191]
	Colon-26 tumor-bearing BALB/c mice, exposed to [^111^In]-DTPA-radiolabeled PEGylated NIOs for biodistribution analysis	Span (20/40/60/80)/ Chol/DCP/ PEG-DSPE; TFH method	Remote chelation of [^111^In] into DTPA-containing PEGylated NIOs using oxine as a carrier	Remote [^111^In] labeling of PEGylated NIOs enabled high labeling efficiency and serum stability; Span 20-based NIOs showed enhanced tumor accumulation, while Span 80-based NIOs exhibited reduced circulation stability and increased spleen uptake	[192]
Chelator-mediated radiolabeling of GSH-loaded NIOs	4T1 tumor-bearing BALB/c mice, exposed to [^99m^Tc]-HMPAO-radio-complexincorporated via GSH-mediated passive diffusion into preformed NIOs	Tween 60/Brij 35/Chol; TFH method	Chelator-mediated encapsulation using [^99m^Tc]-HMPAO; passive bilayer diffusion followed by GSH-driven trapping in the aqueous core	[^99m^Tc]-HMPAO-labeled GSH-NIOs exhibited higher tumor accumulation and prolonged blood circulation vs. free tracer; radiolabeling stability ensured by GSH entrapment; enhanced tumor uptake via EPR effect; successful in vivo tracking by SPECT/CT	[189]

Abbreviations: ACM, Acemetacin; EPR, Enhanced Permeability and Retention Effect; TPGS-DTPA, d-α-tocopherol polyethylene glycol 1000 succinate-diethylenetriaminepentaacetic acid; HMPAO, hexamethylpropyleneamine oxime; NIOs, niosomes; PEG-DSPE, 1,2-distearoyl-sn-glycero-3-phosphoethanolamine-N-[methoxy (polyethylene glycol)-2000]; GSH, glutathione; SPECT/CT, Single Photon Emission Computed Tomography/Computed Tomography; TFH, thin-film hydration.

**Table 10 pharmaceutics-17-01473-t010:** Representative multiple stimuli-responsive niosomal formulations.

Specific Modulations	Biological Model and Therapeutic Agent	NIOs Composition and Preparation	Remarks and Applications	Ref.
Dual pH/thermo-sensitive PEGylated NIOs based on DPPC composition	Breast cancer cells (MCF-7 and BT-474) exposed to *Hedera helix* extract (HHE) or *Glycyrrhiza glabra* extract (GGE) encapsulated in responsive NIOs	Tween 60 or Span 60/Chol/DSPE-PEG/DPPC; TFH method	Dual-stimuli responsive release: enhanced extract release at 42 °C and acidic pH (4.5–5.2); triphasic release behavior (intrinsic pH-sensitivity via pH-cleavable DSPE-PEG and thermo-sensitivity via DPPC phase transition); superior cytotoxicity of NIOs-HHE vs. NIOs-GGE; antibacterial activity confirmed; enhanced cellular uptake with reduced IC_50_ vs. free extracts; upregulation of p53 by 60% and down-regulation of MCL-1 genes by 33%, confirming improved anticancer mechanism	[194]
Dual pH/thermo-sensitive NIOs co-functionalized with FA and trastuzumab	SKBR-3 (HER2^+^ breast cancer cells), treated with gemcitabine and CdSe/ZnS QDots-loaded NIOs	Span 60/Tween 60/Chol/DPPC/DSPE-CA-PEG2000/DSPE–PEG/DSPE-PEG-Maleimide; TFH method	FA (via DSPE–PEG) and trastuzumab (via PEG–Maleimide–thiol linkage) as dual-targeting ligands; DPPC as thermo-sensitive lipid (Tc > 41 °C); DSPE–CA–PEG2000 as pH-sensitive polymer (cleavable in acidic tumor microenvironment); high entrapment efficiency (~96.9%) and stability up to 6 months at 4 °C (EE 94.8%); triggered release up to 92% at 42 °C/pH 6.5; 4.5-fold higher cytotoxicity and >2600-fold cellular uptake in SKBR-3 vs. controls; induced 42% of total apoptosis to the cells via p53 activation and MCL-1 suppression; theranostic potential via CdSe/ZnS QDots	[195]
Dual pH/magnetic-sensitive NIOs incorporating NiCoFe_2_O_4_@Silica nanoparticles	MDA-MB-231 (triple-negative breast cancer), SK-BR-3 (HER2+ breast cancer) and MCF-10A (non-tumorigenic) cells treated with NiCoFe_2_O_4_@Silica@NIO co-loaded with curcumin and letrozole	Magnetic NiCoFe_2_O_4_ core coated with a porous silica shell (reservoir for letrozole), further encapsulated in a Span 80/Chol/DCP niosomal bilayer containing curcumin; TFH method	Magnetic NiCoFe_2_O_4_@Silica@NIOs enabled dual-drug co-delivery: magnetic NiCoFe_2_O_4_ core for guidance and MRI tracking and letrozole–curcumin as synergistic therapeutic payloads (theranostic function); pH-dependent release profile; enhanced cytotoxicity in MDA-MB-231 and SK-BR-3 cells vs. free drugs; reduced migration and apoptosis induction confirmed by gene and flow-cytometry analyses; selective biocompatibility in MCF-10A cells confirmed via low uptake; one of the lowest IC_50_ values reported for this drug combo; apoptosis induction, increased cellular uptake, and significant cell-cycle arrest; migration inhibition (~59% vs. 100% controls); selective biocompatibility in MCF-10A cells; magnetically guided, pH-responsive co-delivery with synergistic anticancer selectivity	[196]

Abbreviations: CdSe/ZnS QDs, cadmium selenide zinc sulfide quantum dots; DSPE-CA-PEG2000, 1,2-distearoyl-sn-glycero-3-phosphoethanolamine–citraconic amide–polyethylene glycol; DSPE-PEG, distearoyl phosphoethanolamine-polyethylene glycol; DPPC, 1,2-dipalmitoyl-sn-glycero-3-phosphocholine phospholipid; DCP, dicetyl phosphate; FA, folic acid; HER2+, human epidermal growth factor receptor 2-positive; NIOs, niosomes; TFH, thin-film hydration.

**Table 11 pharmaceutics-17-01473-t011:** Niosomal Drug Delivery Systems and Their Preclinical/Clinical Evaluation.

Type of Niosomal DDS/Formulation Considerations	Medical Applications	Preclinical/Clinical Tests and Findings	Clinical Trial Number	Ref.
Ab-modified NIOs carrying daunorubicine/Anti-CD123 antibody thiolated with Traut’s reagent, Mal-PEG2000-DSPE obtained by post-insertion method	Acute myeloid leukemia	Ex vivo studies on NB4 & THP-1 cells; THP-1 NOD/SCID xenograft; flow cytometry analysis confirmed precise targeting of daunorubicin to NB4 and THP-1 cell lines, proportional to the density of the antibody on the niosomal surface.	-	[28]
Thermo-responsive PEGylated NIOs carrying doxorubicin/Span 60, Chol, SPC80, DSPE-mPEG, obtained by TFH method	Acute myeloid leukemia	Ex vivo studies on KG-1 cells; thermo-responsive NIOs significantly enhanced the cytotoxic effect of doxorubicin on the leukemia cell line KG-1; through PEG coating and temperature-sensitive properties, the obtained NIOs demonstrated prolonged circulation, superior cellular penetration, and controlled drug release.	-	[197]
pH-responsive NIOs carrying antineoplastic agents/Span 60 : Tween60 : Chol : HD-PAA17 and Cur obtained by TFH method	Lymphoma	Ex vivo studies—a panel of tumor cell lines of different origin namely urinary bladder carcinoma (T-24), and cutaneous T-cell lymphoma (HUT-78 and MJ); pH-sensitive Cur-loaded NIOs demonstrated greater colony formation inhibition and enhanced antineoplastic and anti-inflammatory activity compared to free Cur.	-	[161]
NIOs carrying Triamcinolone Acetonide/Brij 52, Span, Cetrimide, Chol, Oleic Acid, Drug obtained by TFH method	Keloids, joint damage in rheumatoid arthritis and psoriatic arthritis/Transdermal/with iontophoresis	Clinical study by histamine wheal suppression test, clinical trial-initial phase, small number of human subjects; compared to conventional preparations even without the application of non-invasive methods such as iontophoresis	Unknow status	[198]
NIOs propolis oromucodhesive film/Span 60 and Chol, polymer combination (EU-L100, HPMC and PVA) prepared by solvent casting technique	Aphthous ulcers/topical application	Completed clinical trial; NIOs improving local drug penetration and retention, through the NIS, and favored accelerated wound healing.	NCT03615820	[199]
NIOs zinc sulfate/Span 60, Chol TFH method	Leishmaniasis/ topical application	Randomized clinical trial; an increased absorption of the niosomal formulation in the target organ with a therapy efficacy similar to the standard protocol (cryotherapy plus intralesional Glucantime). It is recommended as a second-line treatment.	-	[200]
NIOs/Brij 52, Chol, zinc sulfate prepared by TFH method	Verruca vulgaris/topical application	A triple-blind randomized clinical trial; a significant remission of wart Verruca vulgaris lesions following the administration of a 2% niosomal zinc oxide suspension and cryotherapy compared with 2% conventional zinc sulfate suspension combined with cryotherapy.	-	[201]
NIOs gel carrying anthocyanin complex (AC)/prepared by TFH method	Oral wounds/topical application	A randomized block placebo-controlled double-blind clinical trial; the in vitro results indicated that the administration of NIOs increased cell viability, proliferation, migration and expression of proteins with a role in promoting wound healing; the clinical study results showed that the AC niosome gel significantly accelerated wound healing compared to triamcinolone acetonide gel.	-	[202]
NIOs carrying Dexmedetomidine/Chol, Span 60, DCP, dexmedetomidine obtained by thin TFH method	Postoperative analgesia in pediatric cancer patients/rectal administration	Phase 2/3 clinical trial; improved pain management, a viable alternative to intravenous methods, concerns about absorption variability and patient comfort need addressing.	NCT05340725	[203]
NIOs carrying salbutamol sulfate/Span 60, Chol obtained by reverse-phase evaporation method	Pulmonary disease/Inhalation route	Clinical trial phase I; lack of results obtained	NCT03059017	[204]
NIOs gel carrying melatonin/gel (HPMC, CS, and P407), NIOs (Span 60, Chol, melatonin) obtained by TFH method	Sleep induction/transmucosal administration	Clinical trial phase I/II, status Unknown; transmucosal option for melatonin administration with substantial prolonged systemic delivery.	NCT02845778	[204]
NIOs carrying minoxidil/(Span 60, Chol, Tween 20) obtained by ethanol injection method.	Alopecia areata/topical administration	Without FDA-defined phases; in vitro studies showed that can increase transfollicular, transepidermal absorption, controlled release, improved efficacy and superior safety profile compared to conventional topical minoxidil	NCT05587257	[205]

## Data Availability

The original contributions presented in this study are included in the article. Further inquiries can be directed to the corresponding author.

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
