# Peer review of "Niosomes as Vesicular Carriers: From Formulation Strategies to Stimuli-Responsive Innovative Modulations for Targeted Drug Delivery"

_pharmaceutics, 2025, doi:10.3390/pharmaceutics17111473_

Round 1
Reviewer 1 Report
Comments and Suggestions for Authors
The manuscript provides a comprehensive and well-organized review of niosomes as important vesicular carriers, emphasizing recent advances in stimuli-responsive drug delivery systems. The writing language is clear and scientifically sound. The review is well illustrated, includes numerous relevant and up-to-date references, and follows a clear and logical structure. The authors successfully highlight new trends in formulation strategies, modification, and stimuli-responsive approaches.
However, the clinical applicability of such systems remains uncertain, as no niosome-based formulations have yet reached the pharmaceutical market(?). It would be beneficial if the authors briefly discussed the current status of clinical translation and the major challenges, that limit their practical implementation.
Overall, the work is relevant, informative, and contributes meaningfully to the field of nanomedicine. I recommend acceptance after minor revision.
Author Response
Review Report 1. Comments and Suggestions for Authors
We deeply thank you and are grateful for the time and effort dedicated to reviewing our manuscript. Your suggestions help us to improve the manuscript considerably.
Comments and suggestions: the clinical applicability of such systems remains uncertain, as no niosome-based formulations have yet reached the pharmaceutical market(?). It would be beneficial if the authors briefly discussed the current status of clinical translation and the major challenges, that limit their practical implementation.
In this regard, we have added a new subchapter entitled “Clinical Translation Status of Stimuli-Responsive NIOS and conventional NIOS” with all the niosomal formulations that have been included in clinical trials as well as the few published results of these. We have also addressed the negative considerations that limit their practical implementation.
Please find below our detailed responses to your suggestions and comments, as well as the corresponding revision/ newly added section (which are written in red in the re-submitted/revised manuscript file) and for easier viewing I attach it here:
3.6. Clinical Translation Status of Stimuli-Responsive NIOs and conventional NIOs
Niosomes represent today one of the most promising nanotechnological platforms, proving in numerous experimental studies the ability to increase the stability of active substances, improve their solubility and implicitly bioavailability, reduce systemic toxicity and moreover offer a sustained, prolonged release and a superior penetration of biological barriers. However, despite the significant progress made so far, no NIOs-approved therapy for any disorder has been approved by the Food and Drug Administration (FDA).
Stimuli-responsive NIOs, unlike conventional ones, represent a promising direction especially in anticancer drug delivery systems because it ensures a controlled and localized release of drugs, improved therapeutic targeting, increased anticancer efficacy, but nevertheless clinical translation is still in its infancy, with only a few preclinical experimental studies being conducted and with some examples presented in Table 11. The main negative factors influencing their clinical traceability are: limited scalability, reduced physicochemical stability (aggregation, fusion, and drug leakage over time), complex manufacturing processes which are reflected in high production costs but also reduced reproducibility from one batch to another, reduced stability during sterilization processes but also under certain storage conditions, as well as a potentially immunogenic character.
Tabel 11. Niosomal Drug Delivery Systems and Their Preclinical/Clinical Evaluation
|
Type of Niosomal DDS/ Formulation Considerations |
Medical applications |
Preclinical/Clinical Tests and Findings |
Clinical Trial Number |
Ref. |
|
Ab-modified NIOs carrying daunorubicine/ Anti-CD123 antibody thiolated with Traut's reagent, Mal-PEG2000-DSPE obtained by post-insertion method |
Acute myeloid leukemia |
Ex vivo studies on NB4 & THP-1 cells; THP-1 NOD/SCID xenograft; flow cytometry analysis confirmed precise targeting of daunorubicin to NB4 and THP-1 cell lines, proportional to the density of the antibody on the niosomal surface. |
- |
[28] |
|
Thermo-responsive PEGylated NIOs carrying doxorubicin/Span 60, Chol, SPC80, DSPE-mPEG, obtained by TFH method |
Acute myeloid leukemia |
Ex vivo studies on KG-1 cells; thermo-responsive NIOs significantly enhanced the cytotoxic effect of doxorubicin on the leukemia cell line KG-1; through PEG coating and temperature-sensitive properties, the obtained NIOs demonstrated prolonged circulation, superior cellular penetration, and controlled drug release. |
- |
[197] |
|
pH-responsive NIOs carrying antineoplastic agents/ Span 60 : Tween60 : Chol : HD-PAA17 and Cur obtained by TFH method |
Lymphoma |
Ex vivo studies - a panel of tumor cell lines of different origin namely urinary bladder carcinoma (T-24), and cutaneous T-cell lymphoma (HUT-78 and MJ); pH-sensitive Cur-loaded NIOs demonstrated greater colony formation inhibition and enhanced antineoplastic and anti-inflammatory activity compared to free Cur. |
- |
[161] |
|
NIOs carrying Triamcinolone Acetonide/ Brij 52, Span, Cetrimide, Chol, Oleic Acid, Drug obtained by TFH method |
Keloids, joint damage in rheumatoid arthritis and psoriatic arthritis/ Transdermal /with iontophoresis |
Clinical study by histamine wheal suppression test, clinical trial-initial phase, small number of human subjects; compared to conventional preparations even without the application of non-invasive methods such as iontophoresis |
Unknow status |
[198] |
|
NIOs propolis oromucodhesive film/Span 60 and Chol, polymer combination (EU-L100, HPMC and PVA) prepared by solvent casting technique |
Aphthous ulcers/topical application |
Completed clinical trial; NIOs improving local drug penetration and retention, through the NIS, and favored accelerated wound healing. |
NCT03615820 |
[199] |
|
NIOs zinc sulphate/Span 60, Chol TFH method |
Leishmaniasis/ topical application |
Randomized clinical trial; an increased absorption of the niosomal formulation in the target organ with a therapy efficacy similar to the standard protocol (cryotherapy plus intralesional Glucantime). It is recommended as a second-line treatment. |
- |
[200] |
|
NIOs /Brij 52, Chol, zinc sulfate prepared by TFH method |
Verruca vulgaris/topical application |
A triple-blind randomized clinical trial; a significant remission of wart Verruca vulgaris lesions following the administration of a 2% niosomal zinc oxide suspension and cryotherapy compared with 2% conventional zinc sulfate suspension combined with cryotherapy. |
- |
[201] |
|
NIOs gel carrying anthocyanin complex (AC)/prepared by TFH method |
Oral wounds/ topical application |
A randomized block placebo-controlled double-blind clinical trial; the in vitro results indicated that the administration of NIOs increased cell viability, proliferation, migration and expression of proteins with a role in promoting wound healing; the clinical study results showed that the AC niosome gel significantly accelerated wound healing compared to triamcinolone acetonide gel. |
- |
[202] |
|
NIOs carrying Dexmedetomidine/Chol, Span 60, DCP, dexmedetomidine obtained by thin TFH method |
Postoperative analgesia in pediatric cancer patients/rectal administration |
Phase 2/3 clinical trial; improved pain management, a viable alternative to intravenous methods, concerns about absorption variability and patient comfort need addressing. |
NCT05340725 |
[203] |
|
NIOs carrying salbutamol sulphate/Span 60, Chol obtained by reverse-phase evaporation method |
Pulmonary disease/ Inhalation route |
Clinical trial phase I; lack of results obtained |
NCT03059017 |
[204] |
|
NIOs gel carrying melatonin/ gel (HPMC, CS, and P407), NIOs (Span 60, Chol, melatonin) obtained by TFH method |
Sleep induction/ transmucosal administration |
Clinical trial phase I/II, status Unknown; transmucosal option for melatonin administration with substantial prolonged systemic delivery. |
NCT02845778 |
[204] |
|
NIOs carrying minoxidil/(Span 60, Chol, Tween 20) obtained by ethanol injection method. |
Alopecia areata/ topical administration |
Without FDA-defined phases; in vitro studies showed that can increase transfollicular, transepidermal absorption, controlled release, improved efficacy and superior safety profile compared to conventional topical minoxidil |
NCT05587257 |
[205] |
NIOs modified with anti-CD123 monoclonal antibodies were developed by Liu et al. [28] for targeted delivery of daunorubicin (DNR) to target tumor cells in acute myeloid leukemia. This innovative strategy of cytotoxic drug-monoclonal antibody conjugate incorporated into niosomal delivery system allowed for precise targeting of the drug to malignant cells, a significantly increased cellular uptake of up to 3.3-fold compared to unmodified NIOs demonstrated by flow cytometry, significantly enhancing the antitumor effect compared to standard niosomal formulations.
Bahrami-Banan and colleagues [197] developed an innovative thermos-responsive NIOs system for the controlled release of doxorubicin consisting of phosphatidylcholine (20%), Span 60 (52.5%), Chol (22.5%) and DSPE-PEG2000 (5%) - as a new personalized anticancer therapy. The results demonstrated that this niosomal formulation significantly enhanced the cytotoxic effect of doxorubicin on the leukemia cell line KG-1, confirming that thermos-responsive NIOs represent an efficient and versatile platform for improving therapeutic efficacy and reducing adverse effects associated with conventional chemotherapy.
The pH-sensitive copolymer-modified NIOs developed by Gugleva and colleagues [161], showed a higher Cur release rate in acidic media that can be extrapolated as a feasible approach in cancer therapy. In addition, they showed a higher inhibition rate of colony formation, T-24 cells, compared to the free drug. Thus, pH-sensitive NIOs can be explored as a feasible platform for curcumin targeting.
Regarding clinical studies conducted so far are very limited and include non-stimuli-responsive variants and most of them for local/topical uses. Several topical formulations of NIOs have reached clinical evaluation, such as NIOs carrying Triamcinolone acetonide for possible treatment of keloids, joint damage in rheumatoid arthritis and psoriatic arthritis, with proven superior histamine wheal suppression but with unknown clinical trial progress [198].
In a randomized clinical trial, topical application of niosomal zinc sulfate combined with intralesional Glucantime for cutaneous leishmaniasis demonstrated similar efficacy to standard protocol (cryotherapy plus Glucantime) suggesting it as a second-line therapy [200], but data from subsequent clinical trials are lacking.
A triple-blind randomized clinical trial shows that topical application of NIOs-zinc 2% in combination with cryotherapy significantly improved skin penetration, with significantly faster complete remission of lesions (up to 93.3%) associated with Verruca vulgaris disease, with minimal adverse effects and a reduced recurrence rate compared to conventional zinc formulation associated with cryotherapy but data from subsequent clinical trials are lacking [201].
A new niosomal formulation developed for oral tissue regeneration with topical application was the main subject of a randomized block placebo-controlled double-blind clinical trial [202]. In in vitro and clinical studies, the niosomal formulation for the release of anthocyanins (AC) demonstrated a controlled release, high physicochemical stability, superior permeability compared to conventional gel, support of fibroblast proliferation and migration, as well as the synthesis of collagen, fibronectin and laminin, significantly accelerating the healing process of oral wounds. Although there are no further clinical studies, these results confirm that NIOs are a promising platform for the efficient and safe delivery of bioactive compounds in tissue regeneration therapies.
Beyond these localized clinical applications, several clinical investigations have been conducted and completed, including the mucoadhesive film with NIOs as a carrier system for propolis designed to treat recurrent aphthous ulcers with clinical trial number NCT03615820 [199]. The clinical study demonstrated that treatment with mucoadhesive films with propolis-loaded NIOs resulted in a reduction in ulcer size (after 1-2 days) but also much faster healing compared to placebo.
Another niosomal formulation in advanced clinical trials, phase II/III with clinical trial number NCT05340725 [203], aims to evaluate the efficacy and safety profile of rectally administered dexmedetomidine niosomes in pediatric patients undergoing bone marrow aspiration for biopsy. These dexmedetomide NIOs may improve pain management through their sustained release properties of the active substance with analgesic and sedative properties, which would lead to longer-lasting pain relief. The results of this study have not yet been made public, they are under review by the National Library of Medicine. Three additional niosomal formulations have undergone clinical evaluation, although their results remain unpublished. The first involves NIOs carrying salbutamol sulphate, for which preliminary studies have demonstrated an extended release of up to 20 hours compared with less than 2.5 hours for the free drug. The second is a transmucosal niosomal gel containing melatonin (MLT), which showed a residence time of more than 3 hours in ex vivo studies and, in a clinical study involving 14 healthy volunteers, improved the pharmacokinetic profile of MLT in a dose-proportional manner, enhancing absorption and increasing both Cmax and AUC while prolonging T1/2 [204]. The third clinical investigation assessed the effectiveness of transfollicular penetration of niosomal formulations used as carriers for minoxidil in the treatment of alopecia areata [205].
It can be concluded that the number of niosomal formulations translated into clinical trials remains limited, but nevertheless the fact that some have been initiated and completed indicates that there is translational interest in using niosomal formulations as drug delivery systems. Addressing this gap requires a comprehensive and interdisciplinary approach that includes the standardization of fabrication protocols, extensive safety and biodistribution assessments, and the development of customized regulatory frameworks specifically designed for smart nanocarriers [206]. Future progress will depend on simplifying responsive architectures and implementation of standardized analytical methods for assessing stimuli responsiveness, that factors are essential to enable regulatory approval and facilitating the clinical use of stimuli-responsive NIOs as advanced drug delivery systems.

Reviewer 2 Report
Comments and Suggestions for Authors
This review provides a comprehensive and detailed introduction to the preparation principles, methods, and materials used in niosomes, as well as different responsive stimuli formulations, such as pH , redox gradients, enzymatic activity, temperature, light, ultrasound, magnetic or electric fields, thereby enhancing therapeutic precision. The content structure is complete, and there are some small suggestions as follows:
1. It is suggested to supplement the content of multiple response niosomes preparations, as most of the current articles focus on single response research.
2. The proportion of references in the past 5 years is relatively small in the reference list, and it is recommended to update it.
3. Please modify the layout of the images in Table 1.
Author Response
Review Report 2. Comments and Suggestions for Authors
Thank you very much for taking the time to review this manuscript. We are very grateful for your time and the entire effort you have dedicated to review our manuscript. Your valuable comments and suggestions help us a lot to greatly improve the manuscript. Please find the detailed responses below and the corresponding revisions/corrections written in red in the re-submitted/revised manuscript file.
Comments and suggestions 1: It is suggested to supplement the content of multiple response niosomes preparations, as most of the current articles focus on single response research.
Response 1: Thank you so much for your precious comments and suggestions.
I have added additional information to this section on the latest developments in the development of new multiple response niosomes, additional information that is written in red in the manuscript and for your convenience we have copied them here below and also Table 7 which contains additional information written in red.
“Another recently discovered variant of multistimuli-responsive niosomes is dual loading, as theranostic nanovesicular platforms that integrate both therapeutic and diagnostic functionalities. This is achieved when one of the encapsulated payloads acts as an imaging probe – such as quantum dots (CdSe/ZnS), iron oxide nanoparticles, near-infrared dyes (e.g., indocyanine green), gadolinium complexes, or Fe₃O₄ nanocrystals. Such hybrid systems allow for simultaneous drug delivery and real-time imaging or biodistribution monitoring. The advantage of these theranostic platforms is that multistimulus-responsive NIOSs can be activated under internal (pH, enzymes, redox) or external (temperature, magnetic field, light) stimuli, allowing for “on-demand” drug release and generation of optical or magnetic signals for diagnostic tracking [195].
In cancer therapy, the formulation of theranostic NIOs has been widely applied to enhance treatment efficacy, offering several key advantages: (i) spatiotemporal control of drug release through stimuli-sensitive activation, (ii) minimization of systemic side effects by limiting release to tumor tissue, (iii) real-time monitoring of biodistribution through optical or magnetic feedback from the imaging component, and (iv) the possibility of combined therapeutic modalities, such as chemo-photothermal therapy or chemo-imaging [195,196]. This integrated strategy enhances treatment precision and provides physicians with feedback on drug localization, accumulation, and therapeutic response. These multi-stimulus theranostic nanoplatforms exemplify the transition from conventional drug delivery to image-guided smart nanomedicine.”
Table 10. Representative multiple stimuli-responsive niosomal formulations.
|
Specific modulations |
Biological model and therapeutic agent |
NIOs composition and preparation |
Remarks and applications |
Ref. |
|
Dual pH/thermo-sensitive PEGylated NIOs based on DPPC composition |
Breast cancer cells (MCF-7 and BT-474) exposed to Hedera helix extract (HHE) or Glycyrrhiza glabra extract (GGE) encapsulated in responsive NIOs |
Tween 60 or Span 60/Chol/DSPE-PEG/DPPC; TFH method |
Dual-stimuli responsive release: enhanced extract release at 42 °C and acidic pH (4.5–5.2); triphasic release behavior (intrinsic pH-sensitivity via pH-cleavable DSPE-PEG and thermo-sensitivity via DPPC phase transition); superior cytotoxicity of NIOs-HHE vs. NIOs-GGE; acceptable cellular uptake; antibacterial activity confirmed; enhanced cellular uptake with reduced IC50 vs. free extracts; upregulation of p53 by 60% and down-regulation of MCL-1 genes by 33%, confirming improved anticancer mechanism |
[194] |
|
Dual pH/thermo-sensitive NIOs co-functionalized with FA and trastuzumab |
SKBR-3 (HER2⁺ breast cancer cells), treated with gemcitabine and CdSe/ZnS QDots-loaded NIOs |
Span 60/Tween 60/Chol/DPPC/DSPE-CA-PEG2000/DSPE–PEG/DSPE-PEG-Maleimide; TFH method |
FA (via DSPE–PEG) and trastuzumab (via PEG–Maleimide–thiol linkage) as dual-targeting ligands; DPPC as thermo-sensitive lipid (Tc > 41 °C); DSPE–CA–PEG2000 as pH-sensitive polymer (cleavable in acidic tumor microenvironment); high entrapment efficiency (~96.9%) and stability up to 6 months at 4 °C (EE 94.8%); triggered release up to 92% at 42 °C / pH 6.5; 4.5-fold higher cytotoxicity and >2600-fold cellular uptake in SKBR-3 vs. controls; induced 42% of total apoptosis to the cells via p53 activation and MCL-1 suppression; theranostic potential via CdSe/ZnS QDots |
[195] |
|
Dual pH/magnetic-sensitive NIOs incorporating NiCoFe2O4@Silica nanoparticles |
MDA-MB-231 (triple-negative breast cancer), SK-BR-3 (HER2+ breast cancer) and MCF-10A (non-tumorigenic) cells treated with NiCoFe2O4@Silica@NIO co-loaded with curcumin and letrozole |
Magnetic NiCoFe2O4 core coated with a porous silica shell (reservoir for letrozole), further encapsulated in a Span 80/Chol/DCP niosomal bilayer containing curcumin; TFH method |
Magnetic NiCoFe2O4@Silica@NIOs enabled dual-drug co-delivery: magnetic NiCoFe2O4 core for guidance and MRI tracking and letrozole–curcumin as synergistic therapeutic payloads (theranostic function); pH-dependent release profile; enhanced cytotoxicity in MDA-MB-231 and SK-BR-3 cells vs. free drugs; reduced migration and apoptosis induction confirmed by gene and flow-cytometry analyses; selective biocompatibility in MCF-10A cells confirmed via low uptake; one of the lowest IC50 values reported for this drug combo; apoptosis induction, increased cellular uptake, and significant cell-cycle arrest; migration inhibition (~59% vs. 100% controls); selective biocompatibility in MCF-10A cells; magnetically guided, pH-responsive co-delivery with synergistic anticancer selectivity |
[196] |
Comments and suggestions 2: The proportion of references in the past 5 years is relatively small in the reference list, and it is recommended to update it.
Response 2: Thank you so much for your precious comments.
We have completely reorganized the bibliography by introducing new, up-to-date bibliographic references that support the additional information added to the manuscript, and which can be identified by writing in red.
Comments and suggestions 3: Please modify the layout of the images in Table 1
Response 3: Thank you for pointing this out.
We considered correcting the layout of the image in Table 1 to have increased visibility, and for better identification of these changes I attach the corrected form.
Table 3. Aggregation type of self-assembled amphiphiles correlated with CPP values.
|
CPP value |
Aggregation shape |
Surfactant structural characteristics |
Schematic representation |
|
< 1/2 |
Spherical or cylindrical micelles |
Very bulky head (e.g., ethoxylate group) and fairly small tail length (V < l0 x A0 values) |
|
|
1/2 - 1 |
Geometrical packing (vesicles or flexible bilayers) |
If V ~ l0 x A0 (CPP~1) the fairly symmetrical surfactant tends to be packed into cubic or simple lamellar liquid crystalline (Lα) phases, which when dispersed into water form vesicles |
|
|
>1 |
Inverted micelles |
Cone-shaped surfactant with a very bulky tail and a small head and/or short tail (V > l0 x A0 values) |

Reviewer 3 Report
Comments and Suggestions for Authors
-Figure 1: please clarify that the hydrophilic head and hydrophobic tail compose an entity (please add a small figure) which actually is the surfactant that forms the niosomal bilayer. In the current state head and tails seem like distinguished entities.
-2.2. Formulation Factors Affecting NIOs Characteristics: very good analysis, just some readability issues: please make a table with all the formulation mentioned in the sections with at least the following columns: Formulation; Key Material; Material effect; Reference
-please add at least representative figure for each stimuli-responsive niosomal category, reflecting the effect of their properties or release mechanism or structure or stimuli responsive material etc.
-Any clinical translation of the pH-responsive niosomes and current status of any clinical trial. If there is no such clinical registration, please add some comments/reasons that affect negatively their clinical translation.
Author Response
Review Report 3. Comments and Suggestions for Authors
We sincerely thank you for the time and effort you have given to reviewing our manuscript and for the valuable suggestions you have provided. We also thank you for your careful analysis and recommendations that have contributed to the improvement of the work.
Comments and suggestions 1: Figure 1: please clarify that the hydrophilic head and hydrophobic tail compose an entity (please add a small figure) which actually is the surfactant that forms the niosomal bilayer. In the current state head and tails seem like distinguished entities.
Response 1: Thank you for this valuable remark, we took it into consideration and modified the figure according to the recommendations made, the figure which I included in the revised form of the article and which I attach here for better visualization.
Figure 1. Schematic representation of the NIOs structure.
Comments and suggestions 2: Formulation Factors Affecting NIOs Characteristics: very good analysis, just some readability issues: please make a table with all the formulation mentioned in the sections with at least the following columns: Formulation; Key Material; Material effect; Reference
Response 2: Thank you so much for your precious comments and suggestions. I summarized the information regarding the primary factors of the formulation process that influence the stability of niosomes and introduced them into a new Table 4. That can be found in the revised version of the article written in red but which I am also attaching here for better visualization.
Table 4. Summary of formulation factors and materials influencing the characteristics of NIOs.
|
Formulation Factor |
Key Material |
Observed Effect on NIOs |
Ref. |
|
NISs |
|||
|
Alkyl ether surfactants |
Alkyl glyceryl ethers; Brij series; |
Stable, low-allergenic vesicles; suitable for macromolecule delivery. |
[31,55–57] |
|
Alkyl ester derivatives |
Span series |
Non-toxic, non-irritant; EE ↑ with chain length and saturation (Span 60 > Span 40 > Span 20 > Span 80); longer saturated chains ↑ bilayer rigidity and stability. |
[58–61] |
|
Alkyl ester derivatives |
Tween series |
Suitable for hydrophilic drug encapsulation; EE ↓ with chain length (Tween 20 > Tween 60 > Tween 40 > Tween 80); Tween 80 → gene delivery; Tween 20 → epithelial permeability |
[58,62–63] |
|
Pluronic triblock copolymers |
EO–PO–EO type (poloxamers) |
Improve EE and stability; suitable for injectable and thermo-responsive NIOs; Poloxamer 184 ↓ Chol need in mixed systems. |
[64–68] |
|
Glucosyl alkyl ethers (glucosides & alkyl polyglucosides) |
Myristyl-, cetyl-, stearyl-glucosides; Octyl-/decyl-polyglucosides |
Biodegradable, non-toxic; longer chains → stable vesicles; enhance transdermal and cutaneous drug delivery (e.g., tretinoin). |
[69–71] |
|
Bola & Gemini surfactants |
Bola (α,ω-type) & Gemini dimers |
Bola surfactants: ↑ water solubility, ↑ CMC, ↓ aggregation; Gemini surfactants: ↓ CMC, ↑ micelle stability and solubilization; both → non-toxic and non-hemolytic. |
[72–77] |
|
Fatty alcohols & fatty acids |
Stearyl-, cetyl-, & myristyl-alcohol; Stearic-, palmitic-, myristic-, oleic-, linoleic-, octanoic-, & decanoic-acid |
Form bilayer vesicles affecting size, stability, and EE; fatty alcohols → SUVs with controlled release; fatty acids → larger vesicles, stability pH-dependent (near pKa); mixtures improve bilayer stability. |
[78–80] |
|
Key NIS characteristics |
|||
|
Hydrophilic–Lipophilic Balance |
Span, Tween, Brij series |
HLB 4–8 → stable vesicles, ↑ EE (Span 60, Brij 72); high HLB → larger vesicles, ↓ EE (Tween 80, Brij 76); Chol needed for bilayer stabilization. |
[59,60,61,81,84,85] |
|
Critical Packing Parameter |
- |
Influences aggregation and bilayer assembly: ~ 0.5–1 → vesicles; < 0.5 → micelles; > 1 → inverted micelles. |
[86–88] |
|
Gel–Liquid Transition Temperature |
- |
Higher Tc → rigid, stable vesicles, ↑ EE; Lower Tc → flexible, permeable bilayers; Chol stabilizes bilayer via liquid-ordered phase. |
[89,94–97] |
|
Additive Agents |
|||
|
Membrane Additives |
Chol |
Intercalated in bilayer → ↑ rigidity, Tc, and EE, ↓ permeability; high-HLB surfactants need 30–50 mol % Chol; optimal surfactant/lipid ratio (10–30 mM) improves stability. |
[33–34,52,54,98–99] |
|
Surface additives (charge-inducing agents) |
(–): DCP, phosphatidic acid; (+): stearylamine, stearyl pyridinium chloride |
Provide electrostatic stabilization; ↑ EE, ↓ aggregation; optimal 2.5–5 mol%. |
[4,33,45,85] |
|
Steric Additives |
PEG, CS |
Improve colloidal stability via steric (PEG) or combined steric–electrostatic (CS) mechanisms; ↓ aggregation, ↑ rigidity, ↑ mucoadhesion, and prolong drug release. |
[101–103] |
|
Preparation Conditions |
|||
|
Hydration Temperature |
- |
Above Tc → stable bilayer formation; below Tc → defective vesicles; can affects vesicle morphology (polyhedral ↔ spherical); Chol improves thermal stability. |
[105–106] |
|
Hydration Medium pH |
Phosphate buffer with various pH values |
Acidic pH (5–5.5) enhances EE by favoring unionized APIs forms and better bilayer incorporation. |
[54, 107–109] |
|
Hydration medium volume & time |
- |
Optimal hydration parameters → ↑ EE, stable vesicles; Short time or excess volume → large vesicles, drug leakage. |
[62,98,110] |
|
Other Formulation-Dependent Factors |
|||
|
Characteristics of the Encapsulated APIs |
Hydrophilic vs. lipophilic drugs |
Lipophilic APIs → ↑ EE (bilayer partitioning); Hydrophilic → ↓ EE (aqueous core); PEGylation limits size increase via steric hindrance. |
[45,111–112] |
|
Resistance to osmotic stress |
Hypo-/hypertonic media |
Hypertonic → vesicle shrinkage; hypotonic → swelling and accelerated drug release due to bilayer destabilization. |
[45,98,113] |
Comments and suggestions 3: please add at least representative figure for each stimuli-responsive niosomal category, reflecting the effect of their properties or release mechanism or structure or stimuli responsive material etc.
Response 3: Thank you, your valuable comments and suggestions help us greatly to improve the manuscript considerably. We have created a new figure containing the required information for each stimulus-sensitive niosomal category.
Below you will find the figure created and which is also included in the revised version of the manuscript.
Fig 7. Mechanism of Stimuli-Induced drug release in Functionalized NIOSs
Comments and suggestions 4: Any clinical translation of the pH-responsive niosomes and current status of any clinical trial. If there is no such clinical registration, please add some comments/reasons that affect negatively their clinical translation
Response 4: Thank you very much for your comments and suggestions. In this regard, we have added a new subchapter entitled “Status of Clinical Translation of Stimuli-Responsive and Conventional NIOS”, in which we have included all the niosomal formulations that have been included in clinical trials, as well as the few published results of them. Regarding the clinical translation of the pH-responsive niosomes, these data are not published, there are no ongoing clinical trials, but the negative considerations that limit their practical implementation have been addressed and detailed.
Please find below the newly introduced section in response to your valuable suggestions.
3.6. Clinical Translation Status of Stimuli-Responsive NIOs and conventional NIOs
Niosomes represent today one of the most promising nanotechnological platforms, proving in numerous experimental studies the ability to increase the stability of active substances, improve their solubility and implicitly bioavailability, reduce systemic toxicity and moreover offer a sustained, prolonged release and a superior penetration of biological barriers. However, despite the significant progress made so far, no NIOs-approved therapy for any disorder has been approved by the Food and Drug Administration (FDA).
Stimuli-responsive NIOs, unlike conventional ones, represent a promising direction especially in anticancer drug delivery systems because it ensures a controlled and localized release of drugs, improved therapeutic targeting, increased anticancer efficacy, but nevertheless clinical translation is still in its infancy, with only a few preclinical experimental studies being conducted and with some examples presented in Table 11. The main negative factors influencing their clinical traceability are: limited scalability, reduced physicochemical stability (aggregation, fusion, and drug leakage over time), complex manufacturing processes which are reflected in high production costs but also reduced reproducibility from one batch to another, reduced stability during sterilization processes but also under certain storage conditions, as well as a potentially immunogenic character.
Tabel 11. Niosomal Drug Delivery Systems and Their Preclinical/Clinical Evaluation
|
Type of Niosomal DDS/ Formulation Considerations |
Medical applications |
Preclinical/Clinical Tests and Findings |
Clinical Trial Number |
Ref. |
|
Ab-modified NIOs carrying daunorubicine/ Anti-CD123 antibody thiolated with Traut's reagent, Mal-PEG2000-DSPE obtained by post-insertion method |
Acute myeloid leukemia |
Ex vivo studies on NB4 & THP-1 cells; THP-1 NOD/SCID xenograft; flow cytometry analysis confirmed precise targeting of daunorubicin to NB4 and THP-1 cell lines, proportional to the density of the antibody on the niosomal surface. |
- |
[28] |
|
Thermo-responsive PEGylated NIOs carrying doxorubicin/Span 60, Chol, SPC80, DSPE-mPEG, obtained by TFH method |
Acute myeloid leukemia |
Ex vivo studies on KG-1 cells; thermo-responsive NIOs significantly enhanced the cytotoxic effect of doxorubicin on the leukemia cell line KG-1; through PEG coating and temperature-sensitive properties, the obtained NIOs demonstrated prolonged circulation, superior cellular penetration, and controlled drug release. |
- |
[197] |
|
pH-responsive NIOs carrying antineoplastic agents/ Span 60 : Tween60 : Chol : HD-PAA17 and Cur obtained by TFH method |
Lymphoma |
Ex vivo studies - a panel of tumor cell lines of different origin namely urinary bladder carcinoma (T-24), and cutaneous T-cell lymphoma (HUT-78 and MJ); pH-sensitive Cur-loaded NIOs demonstrated greater colony formation inhibition and enhanced antineoplastic and anti-inflammatory activity compared to free Cur. |
- |
[161] |
|
NIOs carrying Triamcinolone Acetonide/ Brij 52, Span, Cetrimide, Chol, Oleic Acid, Drug obtained by TFH method |
Keloids, joint damage in rheumatoid arthritis and psoriatic arthritis/ Transdermal /with iontophoresis |
Clinical study by histamine wheal suppression test, clinical trial-initial phase, small number of human subjects; compared to conventional preparations even without the application of non-invasive methods such as iontophoresis |
Unknow status |
[198] |
|
NIOs propolis oromucodhesive film/Span 60 and Chol, polymer combination (EU-L100, HPMC and PVA) prepared by solvent casting technique |
Aphthous ulcers/topical application |
Completed clinical trial; NIOs improving local drug penetration and retention, through the NIS, and favored accelerated wound healing. |
NCT03615820 |
[199] |
|
NIOs zinc sulphate/Span 60, Chol TFH method |
Leishmaniasis/ topical application |
Randomized clinical trial; an increased absorption of the niosomal formulation in the target organ with a therapy efficacy similar to the standard protocol (cryotherapy plus intralesional Glucantime). It is recommended as a second-line treatment. |
- |
[200] |
|
NIOs /Brij 52, Chol, zinc sulfate prepared by TFH method |
Verruca vulgaris/topical application |
A triple-blind randomized clinical trial; a significant remission of wart Verruca vulgaris lesions following the administration of a 2% niosomal zinc oxide suspension and cryotherapy compared with 2% conventional zinc sulfate suspension combined with cryotherapy. |
- |
[201] |
|
NIOs gel carrying anthocyanin complex (AC)/prepared by TFH method |
Oral wounds/ topical application |
A randomized block placebo-controlled double-blind clinical trial; the in vitro results indicated that the administration of NIOs increased cell viability, proliferation, migration and expression of proteins with a role in promoting wound healing; the clinical study results showed that the AC niosome gel significantly accelerated wound healing compared to triamcinolone acetonide gel. |
- |
[202] |
|
NIOs carrying Dexmedetomidine/Chol, Span 60, DCP, dexmedetomidine obtained by thin TFH method |
Postoperative analgesia in pediatric cancer patients/rectal administration |
Phase 2/3 clinical trial; improved pain management, a viable alternative to intravenous methods, concerns about absorption variability and patient comfort need addressing. |
NCT05340725 |
[203] |
|
NIOs carrying salbutamol sulphate/Span 60, Chol obtained by reverse-phase evaporation method |
Pulmonary disease/ Inhalation route |
Clinical trial phase I; lack of results obtained |
NCT03059017 |
[204] |
|
NIOs gel carrying melatonin/ gel (HPMC, CS, and P407), NIOs (Span 60, Chol, melatonin) obtained by TFH method |
Sleep induction/ transmucosal administration |
Clinical trial phase I/II, status Unknown; transmucosal option for melatonin administration with substantial prolonged systemic delivery. |
NCT02845778 |
[204] |
|
NIOs carrying minoxidil/(Span 60, Chol, Tween 20) obtained by ethanol injection method. |
Alopecia areata/ topical administration |
Without FDA-defined phases; in vitro studies showed that can increase transfollicular, transepidermal absorption, controlled release, improved efficacy and superior safety profile compared to conventional topical minoxidil |
NCT05587257 |
[205] |
NIOs modified with anti-CD123 monoclonal antibodies were developed by Liu et al. [28] for targeted delivery of daunorubicin (DNR) to target tumor cells in acute myeloid leukemia. This innovative strategy of cytotoxic drug-monoclonal antibody conjugate incorporated into niosomal delivery system allowed for precise targeting of the drug to malignant cells, a significantly increased cellular uptake of up to 3.3-fold compared to unmodified NIOs demonstrated by flow cytometry, significantly enhancing the antitumor effect compared to standard niosomal formulations.
Bahrami-Banan and colleagues [197] developed an innovative thermos-responsive NIOs system for the controlled release of doxorubicin consisting of phosphatidylcholine (20%), Span 60 (52.5%), Chol (22.5%) and DSPE-PEG2000 (5%) - as a new personalized anticancer therapy. The results demonstrated that this niosomal formulation significantly enhanced the cytotoxic effect of doxorubicin on the leukemia cell line KG-1, confirming that thermos-responsive NIOs represent an efficient and versatile platform for improving therapeutic efficacy and reducing adverse effects associated with conventional chemotherapy.
The pH-sensitive copolymer-modified NIOs developed by Gugleva and colleagues [161], showed a higher Cur release rate in acidic media that can be extrapolated as a feasible approach in cancer therapy. In addition, they showed a higher inhibition rate of colony formation, T-24 cells, compared to the free drug. Thus, pH-sensitive NIOs can be explored as a feasible platform for curcumin targeting.
Regarding clinical studies conducted so far are very limited and include non-stimuli-responsive variants and most of them for local/topical uses. Several topical formulations of NIOs have reached clinical evaluation, such as NIOs carrying Triamcinolone acetonide for possible treatment of keloids, joint damage in rheumatoid arthritis and psoriatic arthritis, with proven superior histamine wheal suppression but with unknown clinical trial progress [198].
In a randomized clinical trial, topical application of niosomal zinc sulfate combined with intralesional Glucantime for cutaneous leishmaniasis demonstrated similar efficacy to standard protocol (cryotherapy plus Glucantime) suggesting it as a second-line therapy [200], but data from subsequent clinical trials are lacking.
A triple-blind randomized clinical trial shows that topical application of NIOs-zinc 2% in combination with cryotherapy significantly improved skin penetration, with significantly faster complete remission of lesions (up to 93.3%) associated with Verruca vulgaris disease, with minimal adverse effects and a reduced recurrence rate compared to conventional zinc formulation associated with cryotherapy but data from subsequent clinical trials are lacking [201].
A new niosomal formulation developed for oral tissue regeneration with topical application was the main subject of a randomized block placebo-controlled double-blind clinical trial [202]. In in vitro and clinical studies, the niosomal formulation for the release of anthocyanins (AC) demonstrated a controlled release, high physicochemical stability, superior permeability compared to conventional gel, support of fibroblast proliferation and migration, as well as the synthesis of collagen, fibronectin and laminin, significantly accelerating the healing process of oral wounds. Although there are no further clinical studies, these results confirm that NIOs are a promising platform for the efficient and safe delivery of bioactive compounds in tissue regeneration therapies.
Beyond these localized clinical applications, several clinical investigations have been conducted and completed, including the mucoadhesive film with NIOs as a carrier system for propolis designed to treat recurrent aphthous ulcers with clinical trial number NCT03615820 [199]. The clinical study demonstrated that treatment with mucoadhesive films with propolis-loaded NIOs resulted in a reduction in ulcer size (after 1-2 days) but also much faster healing compared to placebo.
Another niosomal formulation in advanced clinical trials, phase II/III with clinical trial number NCT05340725 [203], aims to evaluate the efficacy and safety profile of rectally administered dexmedetomidine niosomes in pediatric patients undergoing bone marrow aspiration for biopsy. These dexmedetomide NIOs may improve pain management through their sustained release properties of the active substance with analgesic and sedative properties, which would lead to longer-lasting pain relief. The results of this study have not yet been made public, they are under review by the National Library of Medicine. Three additional niosomal formulations have undergone clinical evaluation, although their results remain unpublished. The first involves NIOs carrying salbutamol sulphate, for which preliminary studies have demonstrated an extended release of up to 20 hours compared with less than 2.5 hours for the free drug. The second is a transmucosal niosomal gel containing melatonin (MLT), which showed a residence time of more than 3 hours in ex vivo studies and, in a clinical study involving 14 healthy volunteers, improved the pharmacokinetic profile of MLT in a dose-proportional manner, enhancing absorption and increasing both Cmax and AUC while prolonging T1/2 [204]. The third clinical investigation assessed the effectiveness of transfollicular penetration of niosomal formulations used as carriers for minoxidil in the treatment of alopecia areata [205].
It can be concluded that the number of niosomal formulations translated into clinical trials remains limited, but nevertheless the fact that some have been initiated and completed indicates that there is translational interest in using niosomal formulations as drug delivery systems. Addressing this gap requires a comprehensive and interdisciplinary approach that includes the standardization of fabrication protocols, extensive safety and biodistribution assessments, and the development of customized regulatory frameworks specifically designed for smart nanocarriers [206]. Future progress will depend on simplifying responsive architectures and implementation of standardized analytical methods for assessing stimuli responsiveness, that factors are essential to enable regulatory approval and facilitating the clinical use of stimuli-responsive NIOs as advanced drug delivery systems.

Reviewer 4 Report
Comments and Suggestions for Authors
The manuscript "Niosomes as Vesicular Carriers: From Formulation Strategies to Stimuli-Responsive Innovative Modulations for Targeted Drug Delivery" represents a review in a vibrant interdisciplinary topic of drug delivery systems with colloid particles as nanocarriers. The review is well-organized and shows a step forward to analyzing and summarizing recent publications in this field.
The content of the review fits the scope of Pharmaceutics and the respective topic "Complementary Strategies in Drug Delivery: From Particle Engineering to System Optimization".
The review, however, requires the following improvements and clarifications before further considering for publication:
- Line 53 - the manuscript will benefit from appropriate reference.
- Fig. 1 - is it reproduced with permission, open source or drawn by the authors? Please clarify, if the figure is reproduced. Same for other figures, where applicable.
- Fig. 2 and 3 look more like tables. It is recommended to transform them into respective tables.
- The quality of Fig. 8 should be enhanced in the revised manuscript. Some text is hard to read.
- In general, the graphical content of the manuscript looks rather simplistic for a comprehensive review. The authors are recommended to add 2-3 relevant figures from appropriate papers discussing niosomes and related drug delivery systems by requesting electronic permissions, if necessary. This recommendation is particularly relevant to Section 3.
- Lines 299-311 need clarification. It may be rather misleading that hydrocarbon chains look straight in the gel phase. In the lamellar phase, the tails may indeed by more disordered and also interpenetrate (like they actually do in Fig. 1) and the resulting bilayer thickness may also change. The impact of cholesterol on bilayer thickness is also interesting to discuss.
- Lines 593-608: Microfluidization method. Microfluidics is an advanced technique offering tailored synthesis of uniform size nanocarriers. A variety of manuscripts on drug delivery microfluidics were published in recent years (for example: https://doi.org/10.3390/pharmaceutics17010067, https://doi.org/10.1016/j.molliq.2025.128048, and https://doi.org/10.1186/s12967-024-05160-4 ). It is recommended to add some more relevant references to this review section.
- Fig. 8, segment 3 shows a set of microchannels, which actually separate microdrpoplets and are not directly related to microfluidic synthesis of nanoparticles for drug delivery. Please revise.
- The review will benefit from a brief discussion of preparation methods (section 2.3): which of them are classical, although supported by recent publications and which of them are new.
- Table 3 is rather big and some references (for example, 168, 170, 172) represent manuscripts, which were published 10-15 years ago. Should they be included into this review? Also, what is the significant difference between formulations described in Ref. 164 and 166?
Author Response
Review Report 4. Comments and Suggestions for Authors
Thank you very much for taking the time to review this manuscript. We are very grateful for your time and the entire effort you have dedicated to review our manuscript. Your valuable comments and suggestions help us a lot to greatly improve the manuscript. Please find the detailed responses below and the corresponding revisions/corrections written in red in the re-submitted/revised manuscript file.
Comments and suggestions 1: Line 53 - the manuscript will benefit from appropriate reference.
Response 1: Thank you for this valuable remark, we took it into consideration, I have attached reference number 9, which is also on line 55, because it is information that can be found in this article.
Comments and suggestions 2: Fig. 1 - is it reproduced with permission, open source or drawn by the authors? Please clarify, if the figure is reproduced. Same for other figures, where applicable.
Response 2: This figure is a reproduction of the general figure of liposomes from sciencephotogallery, remade and upgraded with the Canvas program.
Comments and suggestions 3: Fig. 2 and 3 look more like tables. It is recommended to transform them into respective tables.
Response 3: Thank you so much for your precious comments and suggestions. Figures 2 and 3 have been converted into table format and will be renumbered Table 1 and Table 2. Was also attached to the manuscript in red, and for easier viewing please find them attached below.
Table 1. Classification of NIOs types.
|
Classification criteria |
Category |
Representative examples |
Main characteristics / Applications |
|
Functionalized derivatives of NIOs |
Ethosomes |
Ethanol or isopropyl alcohol |
Enhanced transdermal penetration |
|
Bola-surfactant NIOs |
Bola surfactants (α, ω-hexadecyl-bis-(1-aza-18-crown-6)) |
Improved transdermal permeability |
|
|
Transfersomes |
Edge activators |
Ultra-flexible vesicles for membrane penetration |
|
|
Discomes |
Cholesteryl poly-24-oxyethylene ether (Solulan C24) |
Large disc-shaped vesicles for ocular drug delivery |
|
|
Aspasomes |
Ascorbyl palmitate |
Intrinsec antioxidant biological activity |
|
|
Elastic NIOs / Ethoniosomes |
Ethanol or edge activators |
Enhanced flexibility and deformability |
|
|
Polyhedral niosomes |
Hexadecyl diglycerol ether (C16G2) or cholesteryl polyoxyethylene ether |
Non-uniform spherical vesicles with a polygonal or faceted shape (typically 4–12 equal sides) |
|
|
Functionalization purpose |
Conventional NIOs |
No special modifications |
Conventional drug delivery |
|
Stealth NIOs (PEGylated) |
PEG modification |
Prolonged circulation, stealth behavior |
|
|
Targeted NIOs |
Ligand-conjugated (antibody, peptide) |
Targeted delivery to tumor/tissue |
|
|
Stimuli-responsive NIOs |
Respond to pH, temperature, enzyme, or magnetic triggers |
Controlled release and smart response |
|
|
Lamellar structure |
Unilamellar NIOs |
~ 10–100 nm ~ 100–1000 nm |
Small Unilamellar Vesicles (SUVs) Large Unilamellar Vesicles (LUVs) |
|
Multilamellar NIOs (MLVs) |
~500–5000 nm |
- |
|
|
Encapsulated molecule type |
Hydrophilic NIOs |
APIs in aqueous core |
Monotherapy |
|
Hydrophobic NIOs |
APIs in bilayer |
Monotherapy |
|
|
Co-loaded NIOs |
Both hydrophilic and hydrophobic APIs |
Combined therapies |
|
|
Encapsulated agent |
Theranostic NIOs |
Drug & imaging agent |
Dual therapy and real-time tracking |
|
Phytoniosomes |
Plant-derived actives |
|
|
|
Protein/Peptide-loaded NIOs |
Proteins, enzymes or peptides |
Targeted macromolecule delivery |
|
|
Gene-loaded NIOs |
DNA, RNA, siRNA, miRNA |
Gene therapy applications |
|
|
Hormone-loaded NIOs |
Hormones (e.g., insulin, estradiol) |
Controlled endocrine delivery |
|
|
Immunoniosomes |
Antibody- or antigen-conjugated |
Vaccine or immunotherapy systems |
|
|
Preparation method |
Conventional hydration |
Thin-film hydration of surfactants + Chol |
Common lab-scale preparation |
|
Proniosomes |
Dried precursors in powder or gel |
Enhanced stability, reconstitution in situ |
Table 2. Advantages and disadvantages of NIOs.
|
Advantages |
Disadvantages |
|
High EE for both hydrophilic and hydrophobic drugs |
Possible leakage of the entrapped drug |
|
Biocompatible & biodegradable, being composed of non-toxic substances |
Potential formation of niosomal aggregates during different preparation stages |
|
Enhanced oral bioavailability and improved skin permeation |
Risk of accumulation, fusion, or leakage of encapsulated drug in niosomal dispersions |
|
Safe and non-toxic for administration via multiple delivery routes (oral, ocular, transdermal, parenteral etc.) |
Variable encapsulation efficiency, especially for large hydrophilic molecules |
|
Provide protection of encapsulated drugs against enzymatic degradation, oxidation, and other destabilizing processes |
Potential local irritation depending on surfactants or excipients used |
|
Effective carriers for targeted, controlled, and sustained drug delivery |
Requirement of specialized equipment for certain preparation methods |
|
|
Challenges in scaling up from laboratory to industrial production |
Comments and suggestions 4: The quality of Fig. 8 should be enhanced in the revised manuscript. Some text is hard to read.
Response 4: Thank you so much for the suggestion, an optimal visualization of the figures improves the quality of our article, so this Figure has been upgraded and attached to this response to the comment but also in the new version of the article and will receive number 6 after the renumbering of the figures.
Figure 6. Schematic representation of NIOs preparation methods: thin-film hydration (1); freeze – thaw (2); microfluidization (3); “Bubble” method (4); sonication (5); reverse-phase evaporation (6); heating method (7); solvent injection (8); proniosomes technology (9); transmembrane pH gradient (10).
Comments and suggestions 5: In general, the graphical content of the manuscript looks rather simplistic for a comprehensive review. The authors are recommended to add 2-3 relevant figures from appropriate papers discussing niosomes and related drug delivery systems by requesting electronic permissions, if necessary. This recommendation is particularly relevant to Section 3.
Response 5: Thank you, your valuable comments and suggestions help us greatly to improve the manuscript considerably. We have created a new figure containing the necessary information for each category of stimuli-responsive niosomes, which falls under section 3 "The Stimuli-Responsive NIOs - New Approaches for Targeting Drug Delivery"
Below you will find the created figure which is also included in the revised version of the manuscript.
Fig 7. Mechanism of Stimuli-Induced drug release in Functionalized NIOSs
Comments and suggestions 6: Lines 299-311 need clarification. It may be rather misleading that hydrocarbon chains look straight in the gel phase. In the lamellar phase, the tails may indeed by more disordered and also interpenetrate (like they actually do in Fig. 1) and the resulting bilayer thickness may also change. The impact of cholesterol on bilayer thickness is also interesting to discuss.
Response 6: Thank you for the valuable comments that help us improve the quality of the article. I redid the image so that it is visible that the thickness of the layer differs between the three phases depending on the orientation of the hydrocarbon chains and their interpenetration. I have also added new information to highlight the impact of cholesterol on bilayer thickness, information that is written in red in this section and that I attach for better visualization
“The bilayer thickness differs among the three phases due to variations in hydrocarbon chain orientation and packing density. It decreases markedly in the liquid-disordered phase (Lα) as a result of chain disorder and interpenetration, whereas in the liquid-ordered phase (Lo), Chol reduces chain orientation, leading to a thickness comparable to or slightly greater than that of the gel phase (Lβ) [93].”
Figure 4. Schematic representation of phase transitions in amphiphilic bilayers: gel phase (a); liquid-disordered phase (b); liquid-ordered phase (c).
Comments and suggestions 7: Lines 593-608: Microfluidization method. Microfluidics is an advanced technique offering tailored synthesis of uniform size nanocarriers. A variety of manuscripts on drug delivery microfluidics were published in recent years (for example: https://doi.org/10.3390/pharmaceutics17010067, https://doi.org/10.1016/j.molliq.2025.128048, and https://doi.org/10.1186/s12967-024-05160-4 ). It is recommended to add some more relevant references to this review section.
Response 7: Thank you so much for the suggestion, these three references are found in the original form of the manuscript, in order to give value to this section I referred again to the last two manuscripts and I accept as the numbering of the bibliographical references the following numbers 132 and 37, which have been added to the text in red.
Comments and suggestions 8: Fig. 8, segment 3 shows a set of microchannels, which actually separate microdrpoplets and are not directly related to microfluidic synthesis of nanoparticles for drug delivery. Please revise.
Response 8: Thank you for this valuable comment, I have redrawn the figure for better image quality but especially I have modified segment 3 according to the required instructions, in order to avoid confusion, particles are formed in the interaction chamber which then exit through a single channel, without being separated. I attach the corrected Figure, and which received the renumbering 6.
Figure 6. Schematic representation of NIOs preparation methods: thin-film hydration (1); freeze – thaw (2); microfluidization (3); “Bubble” method (4); sonication (5); reverse-phase evaporation (6); heating method (7); solvent injection (8); proniosomes technology (9); transmembrane pH gradient (10).
Comments and suggestions 9: The review will benefit from a brief discussion of preparation methods (section 2.3): which of them are classical, although supported by recent publications and which of them are new.
Response 9: Thank you for this valuable remark, we took it into consideration, thus adding a new paragraph in which we made a classification of new and old methods of niosomes preparation, and for a better visualization I have attached it here
“Among the techniques described, several can be regarded as conventional methods, such as thin-film hydration, reverse-phase evaporation, solvent injection, or sonication, which were initially developed for liposomal systems and subsequently adapted for NIOs. These approaches remain widely employed due to their simplicity and reproducibility, being continuously optimized in recent publications [145]. In contrast, modern and emerging methods such as microfluidization, proniosome technology, supercritical CO₂ method, and ball milling represent newer developments that offer improved scalability, solvent-free processing, and enhanced control over vesicle size and uniformity [144]. The coexistence of both classical and advanced approaches highlights the continuous evolution of NIOs fabrication toward more sustainable and industrially viable processes.”
Comments and suggestions 10: Table 3 is rather big and some references (for example, 168, 170, 172) represent manuscripts, which were published 10-15 years ago. Should they be included into this review? Also, what is the significant difference between formulations described in Ref. 164 and 166?
Response 10: Table 3 is larger and contains older articles from 10-15 years ago because we wanted to include all the information on the development of pH-sensitive niosomes, from the beginning to the latest and most innovative methods of preparation. The references in table 164-166 include the same type of functionalized niosomes, but there was a separation of information as a result of the table being shifted to the next page. In the corrected version of the manuscript, we took into account not to repeat this error. For easier viewing please find them attached below.

Round 2
Reviewer 3 Report
Comments and Suggestions for Authors
All remarks adressed with acurate revision. Acceptance suggestion.
Reviewer 4 Report
Comments and Suggestions for Authors
The authors addressed my comments in the revised manuscript.